

# Mapping of sea ice concentration using the NASA NIMBUS 5 ESMR microwave radiometer data 1972-1977

Wiebke Margitta Kolbe[1,2], Rasmus T. Tonboe[1], and Julienne Stroeve[3,4,5]

[1]National Space Institute, Technical University of Denmark (DTU Space), DK-2800 Lyngby, Denmark
[2]Danish Meteorological Institute (DMI), Copenhagen, Denmark
[3]University of Manitoba, Winnipeg, Canada
[4]University College London (UCL), London, UK
[5]University Colorado, National Snow and Ice Data Center (NSIDC), Boulder, Colorado, USA

**Correspondence:** W. M. Kolbe (wmako@space.dtu.dk)

**Abstract.** The Electrically Scanning Microwave Radiometer (ESMR) instrument on board the NIMBUS 5 satellite was a one channel microwave radiometer measuring the 19.35 GHz horizontally polarised brightness temperature ($T_B$) from Dec. 11, 1972 to May. 16. 1977. The original tape archive data in swath projection have recently been made available online by NASA Goddard Earth Sciences Data and Information Services Center (GES DISC). Even though ESMR was a predecessor of modern multi frequency radiometers, there are still parts of modern processing methodology which can be applied to the data to derive the sea ice extent globally.

Here we have reprocessed the entire data set using a modern processing methodology, that includes implementation of pre-processing filtering, dynamical tie-points, and a radiative transfer model (RTM) together with numerical weather prediction (NWP) for atmospheric correction. We present the one channel sea ice concentration (SIC) algorithm and the model for computing temporally and spatially varying SIC uncertainty estimates. Post-processing steps include re-sampling to daily grids, land-spill-over correction, application of climatological masks, setting of processing flags and estimation of sea ice extent, monthly means and estimation of trends. This sea ice dataset derived from NIMBUS 5 ESMR extends the sea ice record with an important reference from the mid 1970s. To make a consistent analysis of the sea ice development through time easier, the same grid and landmask as for EUMETSAT's OSI-SAF SMMR based sea ice CDR have been used for our ESMR dataset. SIC uncertainties have been included for further ease of comparison to other datasets and time periods.

We find that our sea ice extent in the Arctic and Antarctic in the 1970s is generally higher than those available from the National Snow and Ice Data Center (NSIDC) Distributed Active Archive Center (DAAC) derived from the same ESMR dataset, with mean differences of 240.000 and 590.000 km$^2$, respectively. The largest differences reach up to 2 million km$^2$, when comparing monthly sea ice extents. Such large differences cannot be explained by the different grids and landmasks of the datasets alone, and must therefore also result from the difference in data filtering and algorithms, such as the dynamical tie-points and atmospheric correction.

The new ESMR SIC data set has been released as part of the ESA Climate Change Initiative Programme (ESA CCI) and is publicly available at: http://dx.doi.org/10.5285/34a15b96f1134d9e95b9e486d74e49cf (Tonboe et al., 2023).



## 1 Introduction

Arctic sea ice extent (SIE) in September has been decreasing at a rate of about 12 percent per decade since the launch of modern satellite multi-frequency microwave radiometers in 1978 (Comiso et al., 1980; Tonboe et al., 2016; Onarheim et al., 2018; Stroeve and Notz, 2018; Lavergne et al., 2019). This negative sea ice trend in the Northern Hemisphere already started in the 1970s (Rayner et al., 2003; Walsh et al., 2017), though regional trends can differ, as seen for example within the Barents Sea (Chapman and Walsh, 1991). In the Antarctic there are large regional differences in SIE trends but until recently, the

overall trend was positive due to sea ice dynamics (Turner et al., 2009; Sun and Eisenman, 2021). This has changed however in the last decade as a result of several record lows, and as such overall trends have shifted to a more homogeneous pattern, (Schroeter et al., 2023), and in summer (NDJF) the overall trends are now slightly negative. Until recently, the slightly positive trend was believed to be part of long term natural variability that overshadowed the effects of global warming starting in the 1960s (Wang et al., 2019; Stammerjohn et al., 2008; Thompson and Solomon, 2002; Ferreira et al., 2015; Singh et al., 2019;

Fogt et al., 2022). In order to fully understand the drivers of sea ice variability, extending the sea ice data record backwards in time is essential.

Globally, SIE information prior to the satellite data record was largely from ice charts and ship observations. While there have been efforts to include this data in long-term assessments of sea ice change, the data are typically provided in relatively coarse spatial and temporal resolution (1 deg grid) (Walsh et al., 2019) interpolating in both time and space (Titchner and Rayner,

2014). Only satellite based data-sets offer the ability to cover both hemispheres at improved spatial and temporal resolutions, and generally have consistency in processing methods (Lavergne et al., 2019). Sea ice concentration (SIC) derived from Nimbus 5 Electrically Scanning Microwave Radiometer (ESMR) data was previously processed by Parkinson et al., 2004. Here we apply a new processing method that is comparable to the EUMETSAT/ESA CCI SIC record from 1978 and on-wards (see Andersen et al., 2006; Tonboe et al., 2016). This method reduces atmospheric noise regionally over both ice and water surfaces

and uses the pre-processed data to develop a SIC algorithm calibration that is effective in removing both instrument drift and offsets. Seasonal sea ice signature variations are removed by using dynamical tie-points. Lastly, the algorithm calculates time and spatially varying uncertainty estimates.The ESMR SIC data are presented on the same grid and with the same masking as the EUMETSAT/ ESA CCI record, which makes these two records directly comparable. This and the modern processing chain mentioned above warrant the reprocessing presented in this article.

In the following Section 2, the satellite and reanalysis data are described, including the formatting and initial filtering of the data. Section 3 describes the radiative transfer model (RTM) used for the atmospheric correction, the dynamical tie-points, the SIC algorithm with uncertainty estimations, the land-spill-over method and data flags assigned during post-processing. In Section 4, the resulting SIC dataset is presented and compared to other datasets. Finally, Section 5 consists of a discussion and Section 6 provides the conclusions of this work.





## 2 The NIMBUS 5 ESMR instrument and data

The NIMBUS 5 ESMR instrument was a cross-track scanner measuring at 78 scan positions perpendicular to the flight track with a maximum incidence angle of about 64 degrees to both sides. No direct observations at nadir have been made, the closest positions being at +/- 0.7 degrees. The near circular orbit height was about 1112 km with an inclination of 81 degrees. The phased array antenna dimensions was 85.5 x 83.3 cm and the spatial resolution about 25 km near nadir increasing to about 160 x 45 km at the edges of the swath. The full swath was about 3100 km with varying incidence angle and spatial resolution giving a very good (unprecedented) daily coverage in polar regions with no gaps, i.e. no pole holes. The ESMR onboard the NIMBUS 5 satellite was a one channel 19.35 GHz horizontally polarised microwave radiometer operating from 11. December 1972 until 16. May 1977 (1617 days) with some interruptions (see list of days with missing files in Appendix A2). Due to a hot-load anomaly, there are major data gaps between March to May and again in August 1973. Another major data gap occurred from 3 June 1975 until 14 September 1975 because the ground segment was used for receiving Nimbus 6 data instead. When resuming operation in September 1975 the instrument was only operated approximately every other day. From late 1976 to the end of the mission, operation was highly irregular. The last file in the data-set is from 16 May 1977. The data have recently been made available online by NASA in the original tape archive format (TAP-files).

### 2.1 Formatting and co-location of brightness temperatures and ECMWF ERA5 data

The ESMR data were retrieved from the NASA Goddard Earth Sciences Data and information services center (GES DISC) on-line data archive (NASA GSFC, 2016).This data set contains, along with a number of instrument and geographical parameters, 19.35 GHz calibrated brightness temperatures expressed in units of Kelvin. The raw data was recovered by NASA from the magnetic tapes, called Calibrated Brightness Temperature Tapes (CBTT), where they were stored in the original binary TAP file format, each file corresponding to a particular orbit (NASA GSFC, 2016).

All variables in the TAP files were read using online NASA software and converted to NetCDF format without changing the original data structure. Each data point in the TAP file was matched with European Centre for Medium Range Weather Forecast (ECMWF) ERA5 re-analysis data (Hersbach et al., 2020) in time and space (nearest) and appended to a NetCDF file, serving as input to the processing chain. The resulting data are structured in arrays line by line (across-track). Appendix A1 summarizes the variables included in the NetCDF files.

### 2.2 Initial filtering and correction of brightness temperatures

NASA provides a correction on the brightness temperature data to account for lobe structure, antenna loss and angular $T_B$ variation NASA CR, 1974. According to NASA the correction was needed because: "The cause of the gross variations in antenna properties which were observed soon after launch has been determined to be a cross-polarized grating lobe [. . . ] The problem does not exist for the near-nadir beam positions so these positions are unaffected. [. . . ] An empirical calibration has been developed which removes the effect of the lobe structure and antenna loss, which vary with position, and roughly corrects for angular variations in viewing geometry." (NASA CR, 1974, p.400).



Originally, it was planned to use only lobe corrected $T_B$s with their natural angular dependency, but we did not find a way to extract this in the NASA provided data-set. Essentially, only the combined lobe and angular correction, which is a function of brightness temperature, can be removed altogether from the data NASA provides. Thus, the $T_B$s do not vary as a function

of incidence angle, as would be expected for $T_B$s from the sea surface and sea ice.

Despite the corrections done by NASA, the ESMR data still contain erroneous $T_B$s, scan-lines, sudden jumps in the calibration and other obvious artifacts.

Since the ESMR data contains corrupted data and erroneous scan-lines, filtering is needed before the data can be used for sea ice mapping. The filters that we apply are described in Eqns 1-4. They are applied in the same order as described here and

95 if only a single data point or a couple of scan lines are affected only these data points and scan lines are removed from the file. If the whole file is corrupted then it is deleted.

The first filter, the analog filter, which is used for filtering erroneous $T_B s$ and scanlines, is based on the 16 analog voltage entries in the data. The users guide does not explain very well what the 16 entries really are but jumps in these analog signals correspond to anomalous $T_B s$. Our analog filter computes the absolute gradient in the analog signals and anything over a value

of 10 is removed. This threshold was estimated experimentally.

The following set of filters are applied using the processed $T_B$ (in Kelvin) from the previous step. The filters are applied in the following order:

The second filter in Eq. 1 removes data that are non-physical and outside the expected range for sea and ice surfaces.

$$90K < T_B \leq 273.15K \tag{1}$$

For all data points that lie inside the range specified in Eq. 1, the third filter in Eq. 2 removes erroneous scan-lines (across track rows). The threshold of 50K was estimated experimentally:

$$\frac{\sum_j^n |T_{B_{j,i+1}} - T_{B_{j,i}}|}{n} > 50K \tag{2}$$

where $T_{B_i}$ in Eq. 2 is an across track row of $T_B$ and $i$ is an index along track, while $j$ is an index across track.

The fourth filter in Eq. 3 removes single $T_B$ outliers:

$$|p_i - p_{i-1}| + |p_{i+1} - p_i| \geq 150K \tag{3}$$

where $p$ is a single pixel $T_B$ and $i$ is an along track index. The fifth filter in Eq. 4 removes neighbouring $T_B s$ which are locked on the same value.

$$|p_{i+6} - p_{i+5}| + |p_{i+5} - p_{i+4}| + |p_{i+4} - p_{i+3}| + |p_{i+3} - p_{i+2}| + |p_{i+2} - p_{i+1}| + |p_{i+1} - p_i| \neq 0 \tag{4}$$

The outer data points of the swath edges showed significant higher noise levels and coarser resolution than the near nadir

data points (Veng, 2021). Therefore, after the filtering we additionally remove the 4 outermost data points of the swath, corresponding to incidence angles between 57 and 64 degrees on both sides. The new outer edges of the swath data is then at ~56





degrees, similar to modern microwave radiometers for sea ice retrieval of 50-55 degrees (AMSR (Meisner and Wentz, 2000), SMMR (Wentz, 1983), SSM/I (Wentz, 1997)).

Before the filtering, the dataset contained 13496 orbital data files; after filtering there are 10649 ($\sim$79%) good files left. A
120 complete list of days where data are missing after filtering and no SIC could be retrieved is given in appendix A3.

## 3 The radiative transfer model

The Radiative Transfer Model (RTM) has been developed specifically for atmospheric noise reduction and it is comparable to the RTM's used in Andersen et al., 2006 and Tonboe et al., 2016 but with the addition that this ESMR RTM (Eq. 5) can be applied for different incidence angles over both ocean and ice. The RTM takes as input: atmospheric columnar water vapor
$V$ [mm or kgm$^{-2}$], 10 m wind speed $W$ [ms$^{-1}$], atmospheric columnar cloud liquid water $L$ [mm or kgm$^{-2}$], sea surface temperature $Ts$ [K], ice emitting layer temperature $Ti$ [K], sea ice concentration $c_{ice}$ [0-1], and incidence angle [deg]. In return, it simulates the top-of-the-atmosphere 19.35 GHz $T_B$ at horizontal polarisation.

$$T_B = RTM(V, W, L, T_s, T_i, c_{ice}, \theta) \tag{5}$$

The RTM uses the atmospheric part of the model described in Wentz, 1997 to compute atmospheric emission, transmissivity
and reflectivity at the sea surface (open water) together with added modules for simulating the sea ice emissivity (Fig. 1) and open water reflectivity as a function of incidence angle.

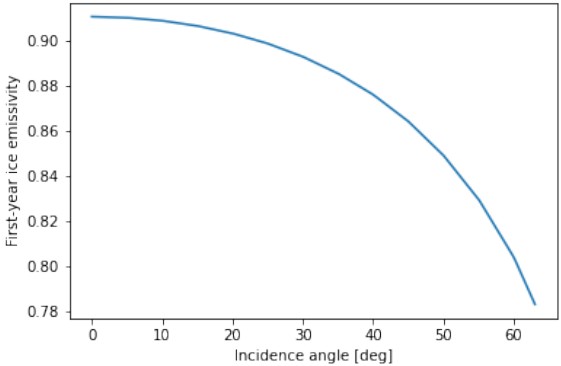

**Figure 1.** The first year-ice emissivity used in the RTM

For the sea ice emissivity, a look-up-table is produced from a simulation using a combined sea ice thermodynamic and emission model during Arctic winter on first-year ice. The thermodynamical and emission model set-up and the simulations are described in Tonboe, 2010. The emissivities as a function of incidence angle are shown in Figure 1 and the look-up-table is
135 given in Table 1.



| Incidence angle, $\theta$, and first-year ice emissivity, $E_{ice}$ | | | | | | | | | | | | | | |
|---|---|---|---|---|---|---|---|---|---|---|---|---|---|---|
| $\theta$ | 0 | 5 | 10 | 15 | 20 | 25 | 30 | 35 | 40 | 45 | 50 | 55 | 60 | 65 |
| $E_{ice}$ | 0.91 | 0.91 | 0.91 | 0.91 | 0.90 | 0.90 | 0.89 | 0.89 | 0.88 | 0.86 | 0.85 | 0.83 | 0.80 | 0.77 |

**Table 1.** Sea ice emissivity look-up-table.

Sea water permittivity, which is used to estimate the calm sea reflectivity (Eq. 6) as a function of temperature, is computed using equation E64 (p. 2046) in Ulaby et al., 1986. The permittivity is almost invariant of the water salinity at 19 GHz and a constant value of 34 ppt is used here.

The calm sea (Fresnel) power reflection coefficient, $r_h$, as a function of the relative permittivity, $\epsilon$, and the incidence angle, $\theta$, for a lossy medium, is computed using Eq. 1.52 in Schanda, 1986, i.e.

$$r_h(\theta) = \frac{(p - cos(\theta))^2 + q^2}{(p + cos(\theta))^2 + q^2}, \tag{6}$$

where

$$p = \frac{1}{\sqrt{(2)}}((\epsilon_r' - sin^2\theta)^2 + \epsilon_r''^2)^{\frac{1}{2}} + (\epsilon_r' - sin^2\theta))^{\frac{1}{2}} \tag{7}$$

and

$$q = \frac{1}{\sqrt{(2)}}((\epsilon_r' - sin^2\theta)^2 + \epsilon_r''^2)^{\frac{1}{2}} - (\epsilon_r' - sin^2\theta))^{\frac{1}{2}} \tag{8}$$

where the relative permittivity $\epsilon_r = \epsilon_r' + \epsilon_r'' j$ of the water surface is a complex number. The calm sea emissivity, $E_0$, is then

$$E_0 = 1 - r_h. \tag{9}$$

The rough water surface emissivity component, $E_W$, is added to the calm sea emissivity, $E_0$, to produce the total sea water emissivity, $E_{water}$. Between ESMR incidence angles of 0 and 63 degrees the 19.35 GHz horizontal polarization $E_W$ sensitivity to wind speed is an almost a linear function ($\frac{\Delta(E_W T_s)}{\Delta W} = 0.0094\theta + 0.3$) of incidence angle, $\theta$, (Meisner and Wentz, 2012), i.e.

$$E_W = \frac{W(0.0094\theta + 0.3)}{T_s}, \tag{10}$$

and

$$E_{water} = E_0 + E_W \tag{11}$$

where $\theta$ is the incidence angle in degrees, $W$ is the wind speed, and $T_s$ is the sea surface temperature [K].

This combination of $E_0$ and $E_W$ follows the procedure described in Wentz, 1997.



The resulting brightness temperature is a linear combination of the sea water and sea ice emission weighted by the SIC following Andersen et al., 2006.

$$T_B = T_{BU} + \tau((1-c_{ice})E_{water}*T_s + (1-c_{ice})(1-E_{water})(\Omega T_{BD}+\tau T_{BC}) + c_{ice}E_{ice}T_i + c_{ice}(1-E_{ice})(T_{BD}+\tau T_{BC})) \quad (12)$$

where $T_{BU}$ is the up-welling brightness temperature from the atmosphere, $\tau$ the atmospheric transmissivity, $E_{water}$ the water surface emissivity, $E_{ice}$ the sea ice emissivity, $\Omega$ the reflection reduction factor due to water surface roughness, $T_{BD}$ the down-welling brightness temperature, and $T_{BC}$ the cosmic background brightness temperature (2.7 K).

EMSR-simulated $T_B s$ and emissivities have been compared with other simulated $T_B s$ using other RTMs for AMSR (Meisner and Wentz, 2000), SMMR (Wentz, 1983), SSM/I (Wentz, 1997) for a constant incidence angle of 55 degrees, which is close to the incidence angle of the other instruments and RTMs (53-55 degrees). The comparison showed that the $T_B s$ of the ESMR RTM are within the same range (approx. 2K) of values of the other models and therefore it seems to be reasonable given the differences in instruments centre frequency and measurement geometry. It is noted that in the correction procedure, differences between two simulated $T_B s$ are used to minimize model biases. Even if the absolute values of the RTM simulated $T_B s$ are biased, this bias would be removed by taking the difference between two simulated $T_B s$, which is the only part used in the correction.

## 3.1 Tie-points and geophysical noise reduction

Tie-points are typical signatures of 100 % ice and open water (0 % ice) and are used in SIC algorithms as a reference for estimating the total ice fraction per satellite pixel $c_{ice}$. Using dynamical tie-points aims to reduce SIC biases that may result from seasonal and inter annual variations of $T_B s$ (Kongoli et al., 2011), instrument drift and RTM and ERA5 biases. For example, Comiso and Zwally, 1980 argue that the variations in average open water $T_B s$ near the ice edge are affected primarily by variations in instrument calibration, and they describe the drop followed by a sharp increase seen in Figure 2 in 1975 as an instrument drift issue.

Our ESMR tie-points are derived on a daily basis from the swath files. Regions of open water and high SICs are selected for each hemisphere, resulting in two regions of sea ice and two of open water for both hemispheres. The selection of the four tie-points is based on criteria set for SIC from ERA5, distance from ice edge, observed brightness temperature, latitude and sea surface temperature, shown in table 2. While computed daily, these are subsequently combined into 15-day running mean tie-points, 7 days ahead and 7 days behind the processed date shown in Figure 2. The 15-day averaging period has been maintained even at the beginning and end of the data-set and when there are data gaps.



|  |  | Ice | Ocean |
|---|---|---|---|
| Arctic |  | 90 >latitude >32<br>sea ice concentration (ERA5) >0.8<br>mean sea ice concentration (ERA5) of<br>    a 5 x 5 grid point box >0.8<br>100 K <brightness temperature <274 K | 90 >latitude >32<br>sea ice concentration (ERA5) = 0<br>mean sea ice concentration (ERA5) of<br>    a 5 x 5 grid point box <0.01<br>sea surface temperature (ERA5) >278 K<br>90 K <brightness temperature <180 K |
| Antarctic |  | -90 <latitude <-48<br>sea ice concentration (ERA5) >0.8<br>mean sea ice concentration (ERA5) of<br>    a 5 x 5 grid point box >0.8<br>100 K <brightness temperature <274 K | -90 <latitude <-48<br>sea ice concentration (ERA5) = 0<br>mean sea ice concentration (ERA5) of<br>    a 5 x 5 grid point box <0.01<br>sea surface temperature (ERA5) >278 K<br>90 K <brightness temperature <180 K |

**Table 2.** Criteria for the four different tie-points.

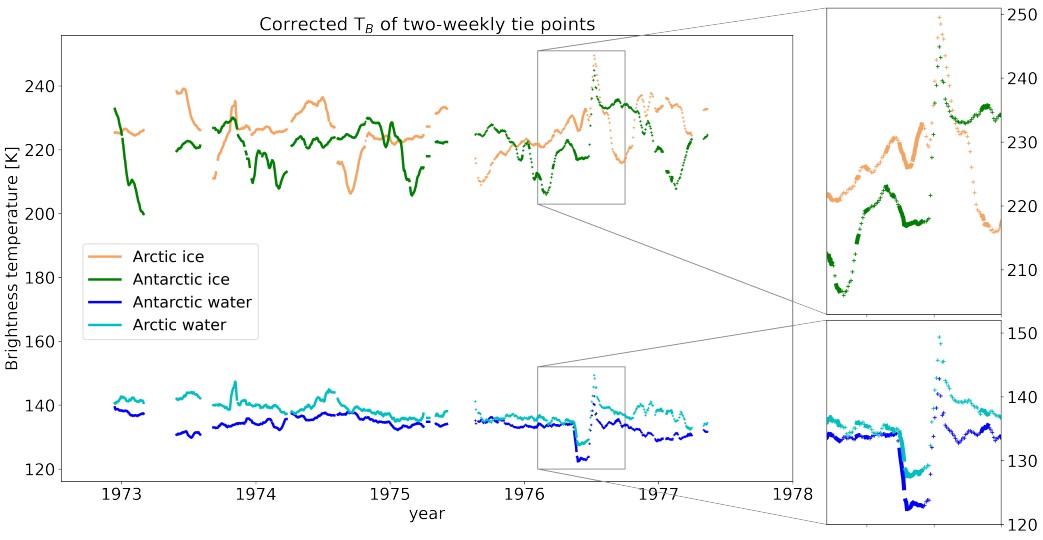

**Figure 2.** The two-weekly tie-points for Arctic and Antarctic ice and water after $T_B$ correction. The boxes are showing the period during May-July 1976 with obvious instrument calibration issues.



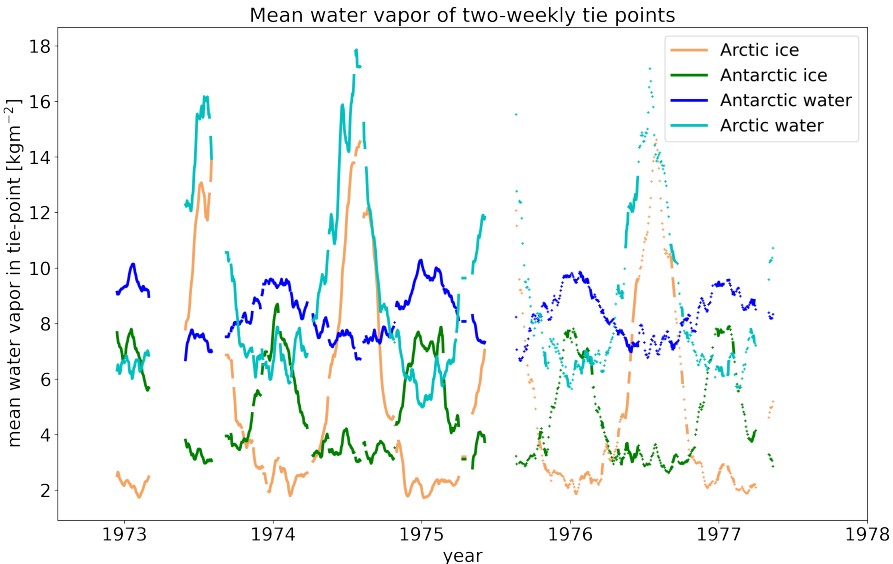

**Figure 3.** Mean Atmospheric water vapor for all grid points included in the four tie-points

The per grid-point $T_B$ correction term $\Delta T_{B,simulated}$, is the difference between a simulated reference $T_B$ using mean values of total column water vapor $[kg/m^2]$ in the atmosphere ($\overline{V}$), 10 m wind speed $[m/s]$ ($\overline{W}$), total column cloud liquid water $[kg/m^2]$ in the atmosphere ($\overline{L}$), the sea surface temperature ($\overline{T_s}$), the ice emitting layer temperature ($\overline{T_i}$) as input to the RTM and a simulated $T_B$ using the actual ERA5 values ($V, W, \overline{L}, T_s, T_i$) for the grid-point. The $T_B$ is not corrected for cloud liquid water, $L$, so the mean $L$ is input to both the reference and the actual simulation. $\Delta T_{B,simulated}$ can both be negative and positive and after correction, the $T_B$s have reduced sensitivity to the geophysical noise sources: $V, W, T_s, T_i$. The fact that the correction term is the difference between two RTM simulations minimizes the impact of biases in the model and the ERA5 data.

The correction term is added to the measured $T_B$, i.e.

$$T_{B,corrected} = T_{B,measured} + \Delta T_{B,simulated}, \tag{13}$$

where

$$\Delta T_{B,simulated} = RTM(\overline{V}, \overline{W}, \overline{L}, \overline{T_s}, \overline{T_i}, c_{ice}, \theta) - RTM(V, W, \overline{L}, T_s, T_i, c_{ice}, \theta) \tag{14}$$

where $c_{ice} = 0$ is the open water tie-point and $c_{ice} = 1$ the ice tie-point. Following (Svendsen et al., 1983), $T_i$ is computed as:

$$T_i = 0.4 \cdot T_{2m} + 0.6 \cdot 272, \tag{15}$$





where $T_{2m}$ is the 2 m air temperature. The horizontal bars above the variable indicate that they are daily mean values for cluster of points selected for the tie-point. The mean water vapor, $V$ in the tie-point is shown in Figure 3.

Figures 4 and 5 show the correction term, $\Delta T_{B,simulated}$, Jan. 1., 1974 over open water in the Northern and Southern Hemisphere respectively. The path-length through the atmosphere is longest at high incidence angles and shortest near nadir, and thus, the absolute value of the correction is largest at high incidence angles. For example, when the atmosphere is driest in the reference compared to the actual simulation, the ends of the corrected $T_B$ turn negative, while they turn strongly positive when the reverse is true.

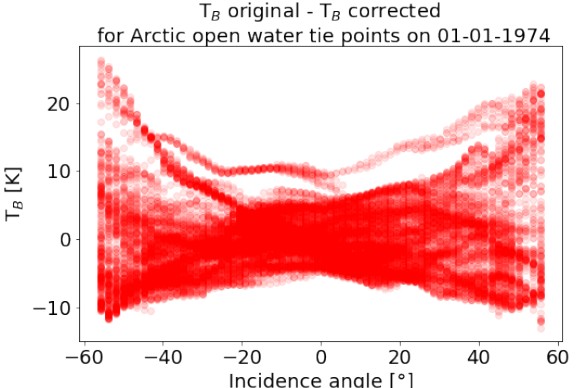

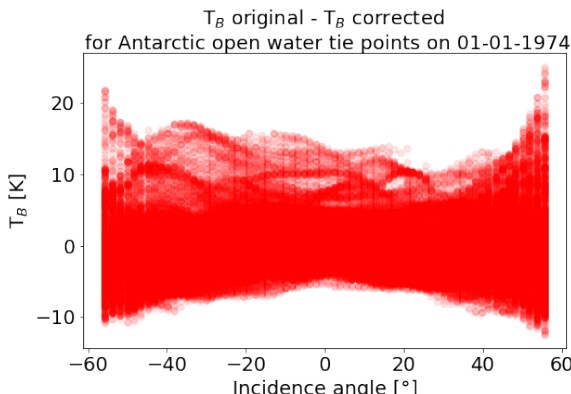

**Figure 4.** Difference of $T_B$s, before and after correction with a mean reference, Northern Hemisphere, open water tiepoints

**Figure 5.** Difference of $T_B$s, before and after correction with a mean reference, Southern Hemisphere, open water tiepoints

The correction works best over open water areas, where it acts as only an atmospheric correction. The RTM appears to better simulate the relevant emission processes in the atmosphere, and the ERA5 data more accurately quantifies the atmospheric noise sources. Over sea ice, geophysical noise sources are related to processes in the snow and ice profile (Tonboe et al., 2021) which are not characterised by the RTM except for the emitting layer temperature $T_i$. The $T_i$, which is used as input to the RTM, is estimated from the 2 meter air temperature in the ERA5 data using Eq. 15. This is important because ESMR is a single channel instrument and thus the $T_B$ and also the derived $c_{ice}$ are sensitive to emitting layer temperature.

The standard deviation of the brightness temperatures for water points in both hemispheres, before and after the correction, is shown in figures 6 & 7.

## 3.2 The sea ice concentration (SIC) and its uncertainty

SIC ($c_{ice}$) is estimated using the measured brightness temperature ($T_{B,measured}$) and the open water ($T_{p,water}$) and ice ($T_{p,ice}$) tie points, i.e.

$$c_{ice} = \frac{T_{B,measured} - T_{p,water}}{T_{p,ice} - T_{p,water}} \tag{16}$$





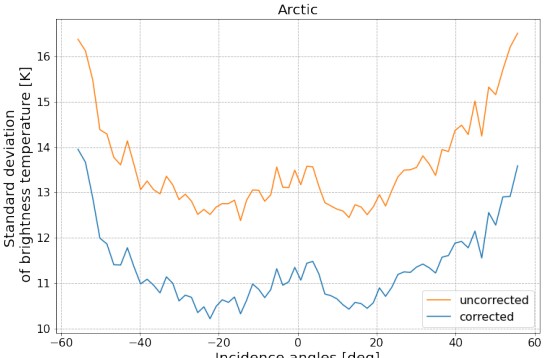

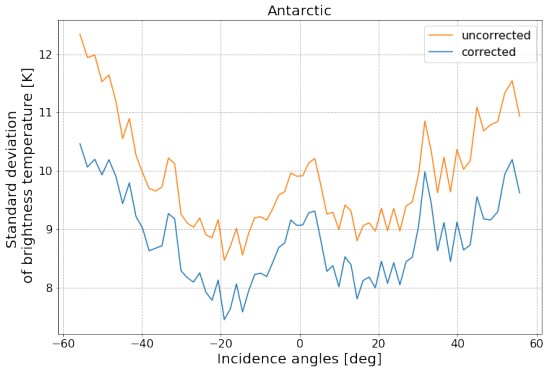

**Figure 6.** Standard deviations of T$_B$s, before and after correction in the Northern Hemisphere, 1974 January. Only filtered for ocean points with ERA5 SIC and SST.

**Figure 7.** Standard deviations of T$_B$s, before and after correction in the Southern Hemisphere, 1974 January. Only filtered for ocean points with ERA5 SIC and SST.

Because the RTM requires $c_{ice}$ as input, $c_{ice}$ is processed iteratively in two steps: 1) the $c_{ice}$ is first estimated using uncor-
rected $T_B$s and tie-points derived from uncorrected data. The $c_{ice}$ estimate is truncated to the interval between 0 and 1 and an
open water filter is applied, forcing all $c_{ice}$ values less than 0.15 to 0. 2) The $c_{ice}$ estimate from step (1) is used in the RTM
calculation (Eq. 5) together with ERA5 data for the geophysical noise reduction of the $T_B$s and $c_{ice}$ is then estimated again in
a second iteration, this time using corrected $T_B$s and corrected tie-points. The mean values of $\overline{V}, \overline{W}, \overline{L}$... used in the reference
simulation is a weighted average with $c_{ice}$ of the mean water and ice tie-point values respectively, i.e. $c_{ice}$ is used as a ratio to
mix the two tie-point values to create mean values of the NWP data for any sea ice concentration.

Iterations to update $c_{ice}$ could in principle continue. However, tests show that updates are small after one iteration and we
only iterate once (e.g. Lavergne et al., 2019).

The total SIC uncertainty is the combination of two components: 1) algorithm uncertainty, which includes instrument noise
and tie-point variability (geophysical noise) and 2) re-sampling uncertainty, which is uncertainty due to data re-sampling.

The algorithm uncertainty is the squared sum of three independent components following Parkinson et al., 1987:

$$\delta c_{ice,algorithm} = ((\frac{\delta T_B}{T_{p,ice} - T_{p,water}})^2 + (\frac{-(1-c_{ice})\delta T_{p,water}}{T_{p,ice} - T_{p,water}})^2 + (\frac{-c_{ice}\delta T_{p,ice}}{T_{p,ice} - T_{p,water}})^2)^{\frac{1}{2}} \qquad (17)$$

where the first term in eq. 17 represents variations due to instrument noise, estimated to a $\delta T_B$ brightness temperature error
of 3 K (Parkinson et al., 1987).

Without the instrument noise term, which is already included in the two tie-point uncertainties, the second and third term in
eq. 17 are used to compute the algorithm uncertainty, $\delta c_{ice,algorithm}$:

$$\delta c_{ice,algorithm} = ((\frac{-(1-c_{ice})\delta T_p, water}{T_{p,ice} - T_{p,water}})^2 + (\frac{-c_{ice}\delta T_{p,ice}}{T_{p,ice} - T_{p,water}})^2)^{\frac{1}{2}} \qquad (18)$$





where $\delta T_{p,water}$, is the water tie-point error, here the (one) standard deviation of the daily tie-point, $\delta T_{p,ice}$, is the ice tie-point error (e.g. one standard deviation of the daily tie-point). The water and ice tie-point errors are weighted by the SIC, and all three errors are normalized with the ice - water brightness temperature contrast and the 2-weekly tie-points. The algorithm

uncertainty is computed on swath data.

The re-sampling uncertainty, $\delta c_{ice,re-sampling}$ is the maximum $c_{ice}$- minimum $c_{ice}$ difference of a 3 x 3 pixel window. The re-sampling uncertainty is computed on re-sampled data (e.g. Lavergne et al., 2019).

The total uncertainty is the squared sum of the algorithm and the re-sampling uncertainty, i.e.

$$\delta c_{ice,total} = (\delta c_{ice,algorithm}^2 + \delta c_{ice,re-sampling}^2)^{\frac{1}{2}} \tag{19}$$

The two uncertainty components and the total uncertainty are included in the data file.

## 3.3 Land-spill-over correction and post-processing

Land-spill-over correction is following the procedure described in Markus and Cavalieri, 2009. A 5 by 5 pixel neighbourhood of the land mask (EASE2 version 2, by OSI-SAF) is analysed to determine which coastal points should be corrected. The land mask is divided into two classes: land points, which are given a value of 90% SIC and open ocean points. If the difference

between the original land mask and the calculated mean mask by the 5 by 5 window is larger than the previously estimated SIC (the RTM corrected & re-sampled SIC), i.e. the SIC is smaller than the theoretical value of the land spill over only, the SIC value is set to 0% and the status flag variable of the data set is raised to 8.

Additionally, a monthly climatology (also by OSI-SAF, same version as land mask) is used to set SIC to 0% and mark open water points by a climatology boundary, which is indicated by a status flag value of 64. Afterwards the land mask is also used

to also mark lakes and coastal areas with status flags 2 and 32 respectively. An overview of all status flag values is shown in table 4:

| status flag values for SIC retrievals | |
|---|---|
| no flag/flag 0 | Nominal retrieval by the SIC algorithm |
| flag 1 | Land |
| flag 2 | Lake |
| flag 4 | SIC is set to zero by the open water filter |
| flag 8 | SIC value is changed for correcting land spill-over effects |
| flag 16 | Handle with caution, the 2m air temperature is high at this position, and this might be false ice |
| flag 32 | Coast |
| flag 64 | SIC is set to zero since position is outside maximum sea ice climatology |
| flag 128 | Point not accepted but no other flags raised |

**Table 4.** Description of the $status\_flag$ variable of the dataset.





The results of the post-processing are included in the daily NetCDF files, for the Northern and Southern Hemisphere, respectively.

## 4 Results

A list of all output variables in the daily SIC files and a short description of them can be seen in Appendix A2. Examples of monthly means of the SIC and mean uncertainty can be seen in figures 8-11.

It is worth noticing that the coverage in figure 8 is complete and because of ESMR's wide swath width of 3100 km and its inclination, the North Pole is covered in contrast to the satellite microwave radiometers following NIMBUS 5 ESMR which have a "pole hole". The area covered by multiyear ice in the central Arctic has lower SIC than the first-year ice regions. This is a consequence of the one channel SIC algorithm. In figure 9 and 11 it can be seen that the uncertainties are largest near the ice edges, as expected. This is due to the re-sampling uncertainty which is dominating near the ice edge where $T_B$ spatial variability is high.

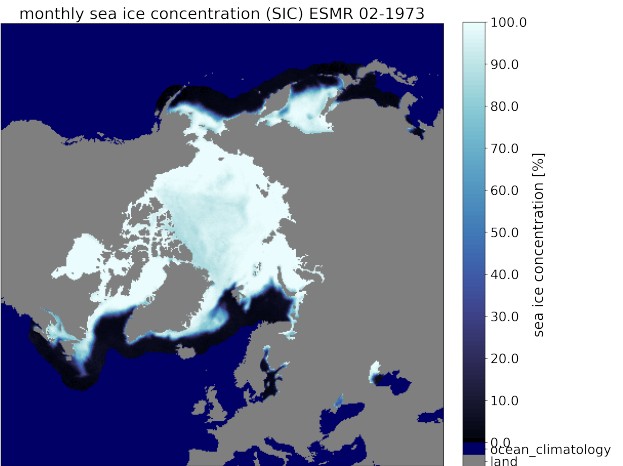

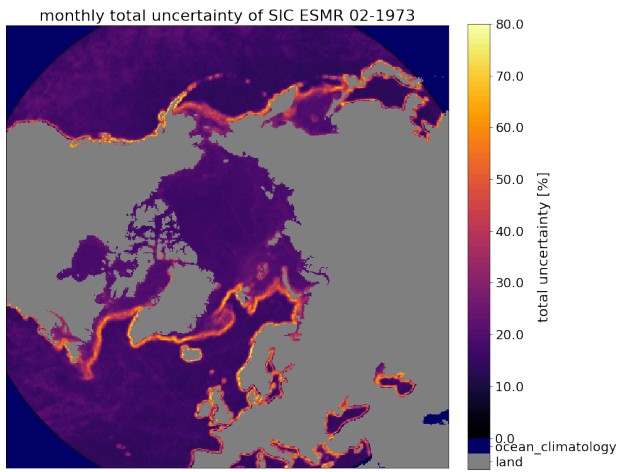

**Figure 8.** Monthly mean SIC for February 1973, Northern Hemisphere.

**Figure 9.** Monthly mean uncertainty for February 1973, Northern Hemisphere. Water areas with no uncertainty due to the ocean climatology are displayed in dark blue.





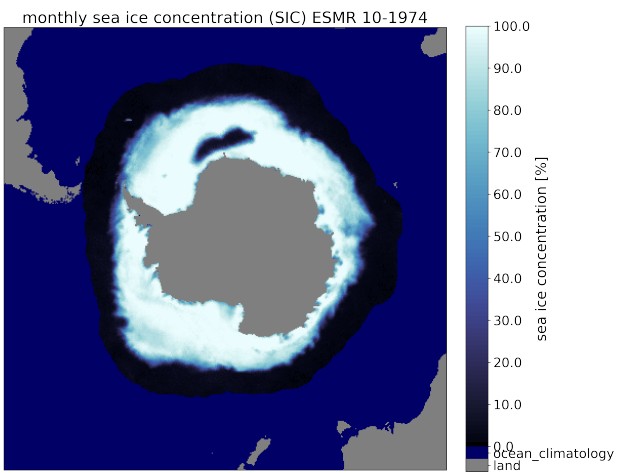

**Figure 10.** Monthly mean SIC for October 1974, Southern Hemisphere

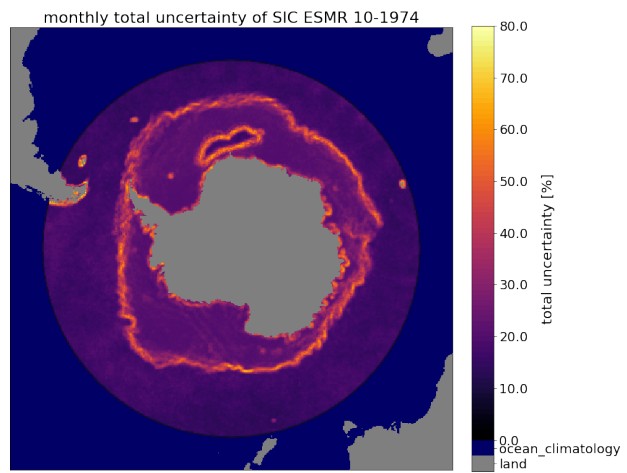

**Figure 11.** Monthly mean uncertainty for October 1974, Southern Hemisphere. Water areas with no uncertainty due to the ocean climatology are displayed in dark blue.

The SIC shows interesting sea ice features in the years 1972-1977. One such feature is the Odden ice tongue extending eastward from the East Greenland Current, visible in figure 8, while another feature is the Maud Rise Polynya, an open water area encircled by sea ice, in the Southern Hemisphere, which can be seen in figure 10. Both examples were much larger in extent in the 1970s and more frequently occurring than they are today (Comiso et al., 2001; Cheon and Gordon, 2019; Jena et al., 2019).

The daily coverage of valid data points that passed all filtering varies a lot through the ESMR operating period. While there is nearly full coverage for the first months, it gets much worse after the summer of 1975 when the instrument only recorded data every second day. An example of the poor coverage is shown for May 1976 in figure 12 and 13.

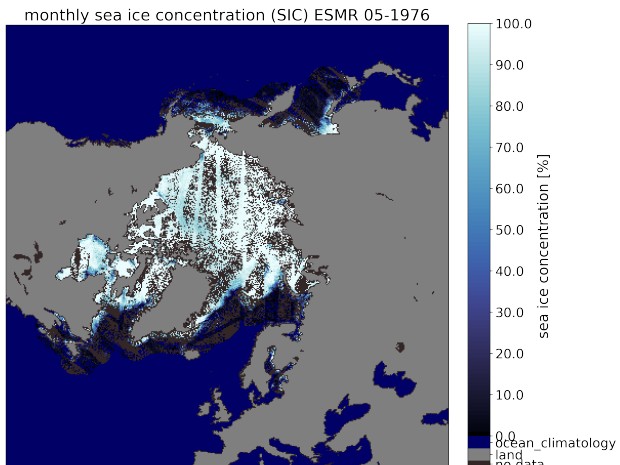

**Figure 12.** Monthly mean SIC for May 1976, Northern Hemisphere

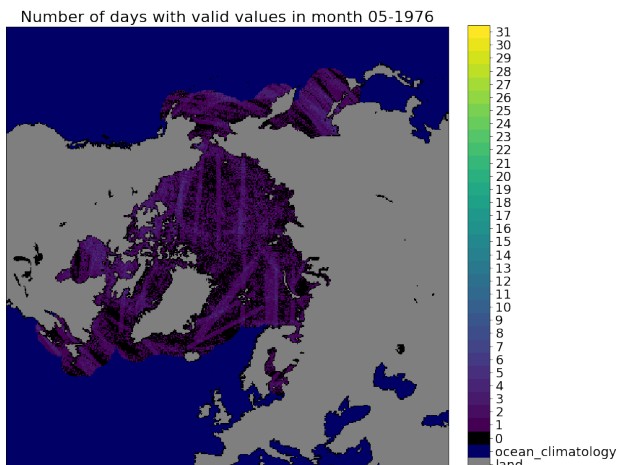

**Figure 13.** Example of poor monthly coverage for May 1976, Northern Hemisphere. Water areas that would show a full daily coverage due to the ocean climatology are displayed in dark blue, to avoid misleading comparisons. Number of days with valid data are indicated by the colorbar.

Monthly averaged SIC are derived to compare our results against other datasets. Only months with a 99% coverage have been used in the comparison, i.e. 99% of all grid points is at least covered once per month. From the monthly SIC, monthly mean SIE are calculated using a threshold of 30% $c_{ice}$. In figures 14 & 15 the ESMR data set (orange line) is shown together with the OSI-SAF CDR (blue line) for 1979-2022 (EUMETSAT, 2017a & EUMETSAT, 2017b) and the sea ice extent derived from NSIDC's NIMBUS 5 ESMR ice concentration product (green line) (Parkinson et al., 2004) using the same threshold for all products (30%).

The comparison shows comparable SIE levels around 1980. In general, our ESMR data set has slightly higher monhtly SIE values than the NSIDC's ESMR product, even though the seasonal pattern is the same.

The mean difference between our sea ice extent and that from NSIDC is 0.24 mill. km$^2$ in the Arctic and 0.59 mill. km$^2$ in the Antarctic for the whole data set.

For the Northern Hemisphere the SIE seems to have been slightly lower during the operational period of NIMBUS 5 ESMR 1972 to 1977 than during the operational period of NIMBUS 7 SMMR from 1978 to 1987. In the Southern Hemisphere the



values of the second half of the 1970s seem to have been around the same magnitude as the largest SIE during the 2014/2015
        season.

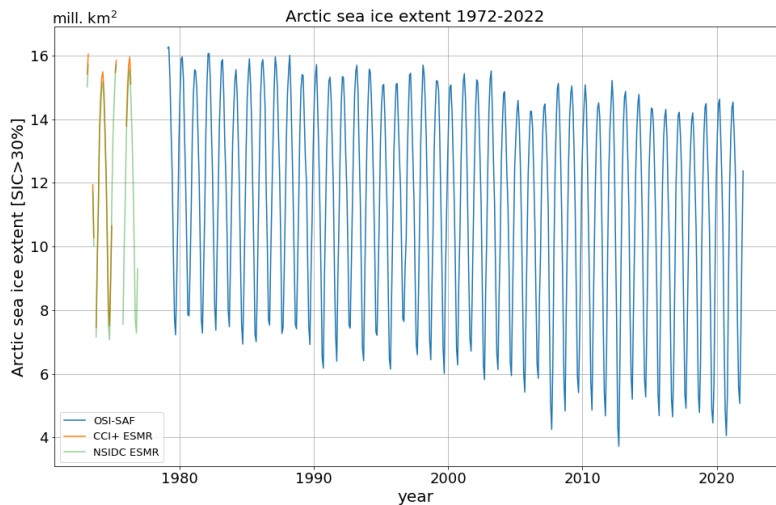

**Figure 14.** Monthly sea ice extent time series for the Arctic based on a 30% sea ice threshold. The orange curve shows values of the ESMR dataset which have a 99% monthly coverage of the hemisphere, while the blue curve is based on the SIC products by OSI-SAF (EUMETSAT, 2017a & EUMETSAT, 2017b), where the 30% threshold has been applied as at: EUMETSAT, 2017c . The green line represents NSIDC's ESMR SIC product. (Parkinson et al., 2004).

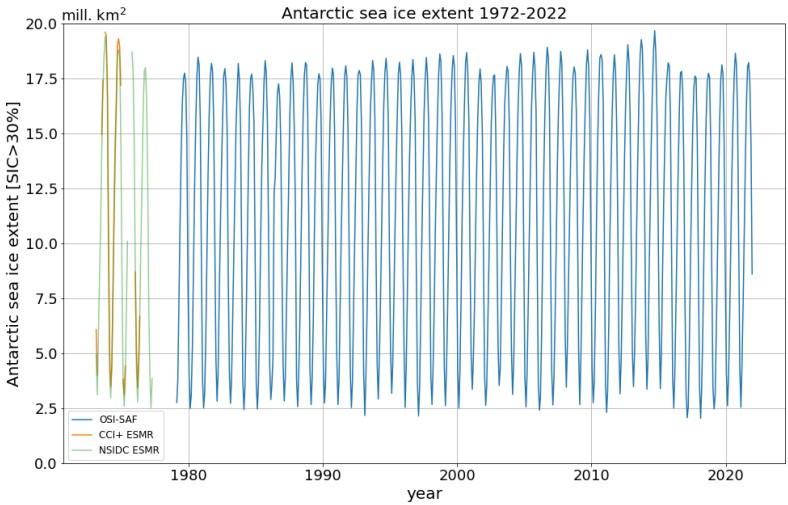

**Figure 15.** Monthly sea ice extent time series for the Antarctic based on a 30% sea ice threshold. The orange curve shows values of the ESMR dataset which have a 99% monthly coverage of the hemisphere, while the blue curve is based on the SIC products by OSI-SAF (EUMETSAT, 2017a & EUMETSAT, 2017b), where the 30% threshold has been applied as at: EUMETSAT, 2017c . The green line represents NSIDC's ESMR SIC product.(Parkinson et al., 2004).

## 5 Discussions

Comparisons between different sea ice products and the new ESMR data set proved to be more difficult than initial expected, since not only the processing algorithms differ, but also the land masks. We were not able to find two independent SIC data 295 sets for 1978 onwards and 1972-77, which share the same land mask.

Thus, it was decided at the beginning of the processing to use the same land mask as the OSI-420 product (1978 onwards) (EUMETSAT, 2020) for our ESMR data set, i.e. a 25 km equal area grid (EASE-2 version 2) land mask, to at least ensure a fair comparison between these two data sets. The NSIDC ESMR data set (green line in figures 14 & 15) used a different land mask with a polar stereographic projection (Parkinson et al., 2004), which also differs from the current NSIDC's CDR land 300 mask.

The difference in SIE is also influenced by the different projections of the data, however, the area difference between the projections is relatively small (only a few thousands of km$^2$), so even a re-projection is expected to yield minimal differences compared to differences caused by the use of different land masks. The comparison of different land masks is complicated by the varying sea ice extent, which exposes more or less land throughout its annual cycle, and thus changes the number of 305 grid-points affected by the land mask.



The land masks land area differ between the OSI-SAF and NSIDC ESMR land mask. A comparison between the land mask land and ocean points between the NSIDC land mask and the OSI-SAF land mask showed a difference for the Northern Hemisphere of 460.000 km$^2$ (north of 60 degrees North), where the NSIDC has more land, while the difference is the opposite and much smaller in the Southern Hemisphere, with only 79.000 km$^2$ (south of 60 degrees South), where OSI-SAF's land

mask has slightly more land. More land points in the land mask result in less available grid points for potential sea ice. The difference in the Southern Hemisphere sea ice extent is significantly larger and opposite to the expected contrast by the land mask difference. In the Northern Hemisphere it is not so clear how much of the SIE differences can be accounted to the land mask or algorithm differences. However, since the SIE differences are varying a lot and the differences in the Southern Hemisphere clearly cannot be explained by the land mask difference alone, it is likely that most of the SIE differences come

from the algorithms and processing methods, such as the atmospheric correction and tie point calculation.

The blue curve in figures 14 & 15 is based on OSI-SAFs SIC products OSI-450 & OSI-430-b (EUMETSAT, 2017b & EUMETSAT, 2017a), corresponding to the SIE of the OSI-420 product (EUMETSAT, 2020), but instead of a 15% threshold, a 30% sea ice threshold has been used that matches the OSI-402-d sea ice extent product (EUMETSAT, 2017c).

The 30% threshold, compared to the more common 15%, was better suited for a comparison between different ESMR SIE

data sets due to the relatively high noise level, which can be seen from the total uncertainty in figures 6 & 7. The uncertainty algorithm has been applied for easier data assessment and comparability to other data sets.

A large amount of ESMR data is currently filtered out, and the 99% threshold for the inclusion to the monthly timeline is especially filtering out the second half of the ESMR data, where large data gaps occurred, as seen in figure 13.

The filters worked as expected and removed erroneous $T_B$s from the raw data. To rescue more data points of the 20% of

ESMR data files, that have been currently filtered out, a reprocessing of the data is planned.

To reduce the uncertainty caused by atmospheric noise, the brightness temperatures were corrected with a RTM using several atmospheric parameters from NWP (ERA-5) data, such as water vapor and wind, as input. This correction showed a consistent reduction of the standard deviation of the brightness temperatures for water points in both hemispheres, as can be seen from figures 6 & 7. Over ice surfaces the correction was less steady, since the RTM is not describing all relevant processes related

to the snow and ice processes, which are the main noise source over sea ice. By correcting for atmospheric effects with ERA-5 data, we might have introduced some noise in the anglular dependency for the SIC, due to the use of an incident angle dependent emissivity in the RTM (figure 1 & table 1).

To avoid biases from the RTM and the NWP data, dynamical tie points have been used, which also calibrate the algorithm to seasonal variations and instrument drift. However, we currently use mean tie-points that are independent of the incident

angle. Therefore, a possible improvement for a future version of the data set might be accomplished by using angle dependent tie-points instead and this will require a complete re-calibration of the NIMBUS 5 ESMR $T_B$ data.

Even after filtering the data for obvious errors it is clear that there are still issues with the absolute calibration of the instrument (Comiso and Zwally, 1980). For example, in 1973 after the hot-load anomaly the ocean $T_B$ in the Southern Hemisphere is several Kelvin below the $T_B$ level before the anomaly and in 1976 there is a dip in May and June followed by a sharp increase



in $T_B$ (Zwally et al., 1983). Low frequency (timescales $\geq$ days) $T_B$ variations and regional variations on hemispherical scales
are compensated by the dynamical SIC algorithm tie-points (Tonboe et al., 2016).

In spite of data gaps and calibration issues, the experimental NIMBUS satellite program was very successful. Applying
modern processing methodologies, including dynamical tie-points and atmospheric noise reduction of the $T_B$s, reduces the
noise over both ice and open water consistently. This newly processed ESMR sea ice data-set extends the existing sea ice
climate data record (CDR) with an important period from the 1970s. This extension of the SIE record contributes to the United
Nations Sustainable Development Goals (SDGs) related to climate change by providing more observations for longer-term
assessments of Arctic and Antarctic sea ice changes.

## 6 Conclusions

In this paper we presented a new SIC data set covering 1972-1977, by using the ESMR data from the Nimbus-5 satellite.
The data set consists of daily netCDF files for the Northern and Southern Hemispheres, respectively. SIC and associated
uncertainties are included in the data set. The uncertainties and choice of same land mask, spatial grid and projection as for
EUMETSATs SIC CDR make comparisons between the time periods easier.

To repeat, the most important findings are:

- While the seasonal pattern is very similar to NSIDC's ESMR SIC product, our product shows systematic larger SIE
  values, which can not be explained by differences between land masks alone. For the Northern Hemisphere our SIE
  values are matching the levels of the 1980s of the OSI-SAF CDR with the same land mask, while values of the Southern
  Hemisphere have been larger in the 1970s than in the 1980s.
- Uncertainty variables have been included in our ESMR data set for better data assessment.
- Atmospheric noise has been reduced with the use of an RTM and the ERA5 atmospheric data.
- Dynamical tie-points were used to avoid biases from the RTM and NWP data as well as to adjust to seasonal variability.

*Data availability.*

The newly processed ESMR data are released through the ESA CCI Open Data Portal:

https://climate.esa.int/en/odp/#/project/sea-ice

DOI: http://dx.doi.org/10.5285/34a15b96f1134d9e95b9e486d74e49cf (Tonboe et al., 2023)



**Appendix A**

**A1**

Table A1: The table is showing the data variables in the processing input NetCDF file and a description of each variable

| Satellite variables | |
|---|---|
| Time | Time of data [year, month, day, hour, minute, second] |
| Brightness_temperature | Brightness temperature of the 78 scan spots [Kelvin x 10] |
| Latitude | latitude of the 78 scan spots [degrees x 10] |
| Longitude | longitude of the 78 scan spots [degrees x 10] |
| Pitch_fine_error | Pitch fine error [degrees x 10] |
| Roll_fine_error | Roll fine error [degrees x 10] |
| RMP_rate | RMP indicated rate high [x 10] |
| NADIR_LAT | Sub-satellite latitude [degrees x 10] |
| NADIR_LON | Sub-satellite longitude [degrees x 10] |
| Height | Satellite height [km] |
| Digital_b | A set of 1 bit status words to indicate the position of each of the command relays (users guide p. 83) |
| Status_indicator_1 | A bit status word |
| Status_indicator_2 | A bit status word |
| Data_source | A bit status word |
| Beam_position | A bit status word |
| PGM_id | Unique identification number assigned to program that prepared tapes |
| HOT_MEAN | Hot load mean [x 10] |
| HOT_RMS | RMS of hot load [x 100] |
| COLD_MEAN | Cold load mean [x 10] |
| COLD_RMS | RMS of cold load [x 100] |
| MUX_1 | Average antenna temperature |
| MUX_2 | Average phase shifter temperature |
| MUX_3 | Ferrite switch temperature |
| MUX_4 | Ambient load temperature |
| MUX_5 | Reference load temperature |
| MUX_6 | Automatic Gain Control |
| Analog_0 | Analog signals (voltages) |
| Analog_1 | Analog signals (voltages) |
| ... | ... |
| Analog_15 | Analog signals (voltages) |





| ERA5 variables | |
|---|---|
| u10 | u component of the wind speed at 10 m (parallel to longitude) $[ms^{-1}]$ |
| v10 | v component of the wind speed at 10 m (parallel to longitude) $[ms^{-1}]$ |
| t2m | 2 m air temperature [K] |
| istl1 | Ice internal temperature [K] |
| ... | ... |
| istl4 | Ice internal temperature [K] |
| lsm | Land-sea-mask |
| msl | Mean sea level pressure [hPa] |
| siconc | Sea ice concentration [0-1] |
| sst | Sea surface temperature [K] |
| skt | Skin temperature [K] |
| tcw | Total column water $[kgm^{-2}]$ |
| tcwv | Total column water vapor $[kgm^{-2}]$ |
| era_time | Valid time for analysis |

Table A2: The table shows the output variables stored in the daily NetCDF files and a description of each variable

| Output variables | |
|---|---|
| ice_conc | filtered sea ice concentration using atmospheric correction of brightness temperatures and open water filters [%] |
| raw_ice_conc_values | raw sea ice concentration estimates as retrieved by the algorithm [%] |
| total_standard_error | total uncertainty (one standard deviation) of sea ice concentration [%] |
| smearing_standard_error | smearing uncertainty of sea ice concentration [%] |
| algorithm_standard_error | algorithm uncertainty of sea ice concentration [%] |
| status_flag | status flag bits for the sea ice concentration as described in table 4 |
| Tb_corr | corrected brightness temperatures [K] |
| Tb | uncorrected brightness temperatures [K] |
| time | Time of data [year, month, day] |
| xc | x coordinate of projection [km] |
| xy | y coordinate of projection [km] |
| lat | Latitude [degrees] |
| lon | Longitude [degrees] |



**Table A3.** The table is showing the missing dates

Missing dates

| Year | Jan | Feb | Mar | Apr | May | Jun |
|---|---|---|---|---|---|---|
| 1972 | | | | | | |
| 1973 | - | 28 | 1-31 | 1-30 | 1-27 | - |
| 1974 | 12-15 | 10-11 | 24-31 | 1-5 | 14-15 | - |
| 1975 | - | 21 | 24,31 | 1-8,16-30 | 1-2 | 3-30 |
| 1976 | 1,3,5,7, 9,13,15,17, 19,21,23,27 | 2,6,12, 14,16,18, 22,24,26,28 | 1,3,5,7,9, 11,13,15,19, 21,23,25,27,31 | 2,4,6,8,10, 12,14,16,18,20, 22,24,28,30 | 2,6,8, 10,12,20, 24,30 | 13,15,17, 19,21,23, 25,27,29 |
| 1977 | 9-18,23, 25,27,29,31 | 4,6-8,10, 12,14,16,18, 20-22,24,26,28 | 4,6,8,10, 12,14,16,20, 26,28,30-31 | 1-29 | 1,3,5,7,9,11-16 | |

| Year | Jul | Aug | Sep | Oct | Nov | Dec |
|---|---|---|---|---|---|---|
| 1972 | | | | | | 31 |
| 1973 | 27-28 | 1-31 | 1-4,13-16,22-23 | 14-15 | 9-10,29-30 | 1,5,14-17 |
| 1974 | 17-19 | 1-8,13-14 | - | 22-23,25-27 | 1 | - |
| 1975 | 1-31 | 1-17,19,21-25, 27,29,31 | 5-6,8,10, 12,14,16,18, 20,22,24, 26,28,30 | 2,4,6-8, 10,12,14,18, 20,22,24,26 | 1,3,5,7,9,11, 13,15,17,19, 21,23,25,27,29 | 1,3,5,7,9,11, 13,15,17,19,21, 23,25,27,29,31 |
| 1976 | 1,3,5,7,9, 11,13,15,17,19,21, 23,25,27,29,31 | 2,4,6,8,10, 12,14,16,18,20, 22,24,26,28,30 | 1,5,11, 13,15,17, 23,25,27 | 1,3,5,7,9, 13,15,17,19,21, 23,25,27,29,31 | 2,4,6,8, 10,12,14,16, 18,20,22,24, 26,28,30 | 2,4,6,8, 10,12,14-18, 20,22,24,31 |
| 1977 | | | | | | |



*Author contributions.*  WK performed the experiments, investigated the data, developed software and wrote the manuscript with contributions from all authors. RT contributed to the conceptualization, methodology, software, investigation, original draft and review & editing of the manuscript, and provided supervision throughout the work. JS contributed with critical feedback that shaped the interpretation and presentation of the data and improved the manuscript through review & editing.

*Competing interests.*  The authors declare no competing interests relevant to the work presented in this paper.

*Acknowledgements.*  This work is part of the ESA Climate Change Initiative Programme (ESA CCI) Sea Ice CCI (Sea_Ice_cci) project and the Danish National Centre for Climate Research (NCKF) at the Danish Meteorological Institute (DMI). The authors would like to thank Tadea Veng, Roberto Saldo, Thomas Lavergne, Atle Sørensen and Leif Toudal Pedersen for their inputs and valuable feedback. We would also like to thank Imke Sievers, John Andrew Dawson, and Andreas Svejgaard Jensen for developing and testing the data filters described in section 2.



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
