# Peer review of "Mapping of sea ice concentration using the NASA NIMBUS 5 ESMR microwave radiometer data 1972-1977"

_Earth System Science Data, 2023_

## Referee Comment (RC2)

**Review on "Mapping of sea ice concentration using the NASA NIMBUS 5 ESMR microwave radiometer data 1972-1977" by Kolbe et al. (2023)**

*Submitted to Earth System Science Data (ESSD)*

*13 November 2023*

The submitted paper presents a newly developed sea ice concentration algorithm, that makes use of historic Nimbus-5 ESMR satellite data – a single (low) frequency passive microwave sensor that operated between 1972 and 1977. The new re-processed data set is intended to extend previous CDR of passive microwave sea ice observations back in time. Not only due to an often quite low data quality, with prolonged data gaps especially towards the end of the covered time period, a range of modern processing techniques, various filters steps and additional post-processing are applied to ensure a sufficient output data quality with added uncertainty quantifications. While differences to sea ice extent (SIE) estimations from other comparable OSISAF CDR seem to be rather low, i.e., showing similar magnitudes in the 70ies and 80ies, the authors note apparent differences (positive offset) when comparing this new data set to the older NASA ESMR SIC processing by Parkinson et al. (2004). While land mask differences seem to play a role in that context, the overall SIE differences cannot be explained to full extent at this stage, but seem to be more related to algorithm and processing issues.

The paper is nicely written and, in most parts, well-structured and easy to follow. Below, I list several parts that could benefit from some rephrasing/clarification under general and specific comments. Figures and Tables are generally good and informative, but could here and there be improved by some small tweaks, additions and potentially merging (see technical comments below).

Overall, I consider the study well worked out, so that some rather minor changes could well be sufficient to grant publication.

**General comments**

- I noted several suggestions to make the figures a bit more concise below under "technical comments". While often a stylistic choice, I would consider this to be an area where the paper could easily be improved a bit – also with regard to some descriptions/discussions in sections 4 & 5.
- Please pay attention with your equations - sometimes they miss proper explanations (see specific comments below) and are quite briefly "rushed" over.
- In your "Sea Ice Concentration Product User Guide for ESMR", I noticed a processing flow-chart that didn't make the jump to this paper here. I would consider this to be a nice addition here as well.
- It would have been nice to read a bit more about the core differences between your new approach and the study by Parkinson et al. (2004). For instance, right in the introduction where it is currently just one short sentence. It doesn't need to be overly long, but it might help to get a sense on where algorithm-related differences might originate from, without having to read the full Parkinson et al. study first.
- As already mentioned in RC1, I also got the impression that sub-sections 3.1 to 3.3 do not necessarily relate to a section entitled "The radiative transfer model". Hence, I agree to think about an alternative way of naming / subdividing section 3.

**Specific comments/questions**

**Introduction (Sect.1)**

L.26: I'm sure the references are picked on purpose here, but it seems like a rather long list for this single statement? Is it worth to point out the key differences among those studies here?

**Instrument & Data (Sect.2)**

L.56-68: Reference(s) missing for all these platform / sensor specific information.

L.77: "…and appended to a NetCDF file, …" – it is the same file as the satellite parameters, this could be phrased a bit clearer.

L.107 (Eq.2): "n" not explained in the following

L.110 (Eq.3): Was the threshold for Tb outliers (150 K) also experimentally estimated or was it chosen arbitrarily?

L.111 / Eq.4: There are some explanations for this equation missing (size of the search window & effect of varying it; meaning of $\neq 0$)

**Radiative transfer model (Sect.3)**

L.145 (Eq.7): Careful with the second use of "p". Although different from earlier uses as "p_i" as single pixel indicators, it might be confusing for the reader and should, for instance, be augmented by an additional explanation of "p" and "q" in Eq.7.

L.182: Please provide more details on the ERA5 OSTIA SIC, as they are quite crucial for the selection of tie points (affecting SIC thresholds and presumably also distance to the ice edge). Further, how is the distance to the ice edge defined, and where/how is this criterion depicted in Table 2?

Fig.3: Add data source to caption.

L.200: Add data source for T2m data.

**Results (Sect.4)**

L.267: "This is a consequence of the one channel SIC algorithm." – Can you comment more on this effect? Is there any way / idea to correct for this?

L.267/268: Uncertainties are also high along the coastal margins, especially at lower latitudes. Might be worth to mention and explain.

L.270-272: As noted under technical comments below - hard to locate for readers that are unfamiliar with the geographical setting of one or the other hemisphere. Simple lat/lon indications could help (when present on the maps), or alternatively, a regional close-up figure (daily level? Could then even feature other parameters from the data set, such as uncorrected/corrected brightness temperatures).

L288: "For the Northern Hemisphere the SIE seems to have been slightly lower during the operational period of NIMBUS 5 ESMR 1972 to 1977 than during the operational period of NIMBUS 7 SMMR from 1978 to 1987." – To me & purely based on Fig.14, max. values in winter seem to be more or less similar, while the two available min. values in summer are seemingly even slightly higher than in the 80ies. Am I wrong?

**Discussions (Sect.5)**

L.300: Is it possible to illustrate this with an example? I.e., where do the land masks differ the most?

L.325: You mention a planned reprocessing of some sort to increase the number of rescued data points. Do you already have concrete ideas on how you intend to do that?

L.330: Can you name some examples for "related snow and ice processes" that are causing the noise over sea ice? How about atmospheric effects over sea ice that also trouble other widely used SIC algorithms?

**Technical comments**

L.82: Brackets around reference missing (NASA CR, 1974)

L.100: "a value of 10" – unit missing

Figures 2, 3, 4 & 5: Similar to other line plots, the addition of grid lines could help to make out differences more easily.

Figure 4 & 5: Smaller points could help to reduce the large red "cluster-patches". Further, the captions read more like notes, this could be improved by using proper phrases. Lastly, Fig.4 & 5 could be combined into one Figure 4 (a & b), just to spare two almost identical captions next to each other.

Figure 6 & 7: As before - could be combined into one Figure (a & b) to spare two almost identical captions next to each other.

L.219-225: Steps 1) and 2) could be formatted as bullet points, thereby reducing potential confusions with the numbers just before point (2).

L.254-255: Double use of "also".

Figures 8-13: Multiple comments/suggestions

- Size of individual panels could be increased
- Geographic references are missing in all panels – e.g., thin lat/lon lines could be added
- Is the wide spatial extent of the maps chosen on purpose or would it make sense to zoom in a bit, sparing some lower latitude regions?
- Monthly mean SIC & uncertainties could be combined in merged Figures (a & b)

L.280 & 283: "threshold of 30%" – better write s.th. like "$c\_ice \geq 30\%$"

Figure 14 & 15: As before - could be combined into one Figure (a & b) to spare two almost identical captions next to each other. Plus: be more specific with the thresholds: "… a 30% sea ice concentration (SIC) threshold…" and further "…, where the 30% SIC-threshold has been applied".

L.348 / Section 6: reads more like a Summary and is quite short. One could think about merging this part with section 5 as a "Discussions & summary" chapter, but this is certainly personal taste.

Table A1 (Appendix A): Under "ERA5 variables", I think there is an error with u10m / v10m and their latitude / longitude reference.

Dataset entry on CEDA archive: The document "Algorithm Theoretical Basis Document (ATBD) - ESMR Sea Ice Concentration" is currently not accessible from the ESA website. Has it been moved?

---

## Author Comment (AC1)

**Reply to Referee Comment RC1**

**(review in black, author response in blue)**

**Review of "Mapping of sea ice concentration using the NASA NIMBUS 5 ESMR microwave radiometer data 1972-1977" by Kolbe et al.**

**Summary**

This paper presents a new sea ice concentration product from the Nimbus-5 ESMR sensor that operated from 1972 to 1977. Earlier sea ice products have been created from ESMR, but the sensor is under-utilized because of limited data quality and substantial amounts of missing data, plus the fact that it is a single-channel sensor while subsequent sensors are multi-channel and there is no overlap between them. The method here uses several filters to remove bad data, implements dynamical tie-points and uses a radiative transfer model with NWP data for atmospheric correction. Comparisons show good consistency with the NSIDC ESMR data product, but with extent values that are generally higher due to the different methods.

**General Comment**

This is a valuable new dataset that makes the ESMR data more useful for long-term timeseries analysis and extends the passive microwave sea ice record another 7 years. The filtering methods are well thought out and remove much of the bad data and the RTM with NWP help correct weather effects. It is clear that great care went into the product and that the output is as good as can be expected given the nature of the data. The comparisons with the NASA/NSIDC product are useful and highlights key differences in the methodologies. The paper is well-written and thorough. After revision in response to the very minor comments below, this is acceptable for publication.

The authors would like to thank the referee for reviewing the manuscript and providing constructive feedback.

**Specific comments (by line number):** I think of science as a process of gaining knowledge through the accumulation of evidence to continually get closer to knowing the truth.

95-113: The filter discussion is a bit jumbled in my view. In Line 97, "The first filter" is mentioned, but then in Line 101, it says "The following set of filters are applied…", which is then described as "the second filter" even though it reads to some degree that it is the first filter. I guess I see that the first one is a bit different, but I would just say something like, "The following filters are used…" and then go one-by-one. Or if you want to separate out the analog filter, maybe say something like, "An initial analog filter is used…" and then say something like, "Next, several other filters are employed…."

This is a good point, and we agree that this section needs revision. Since we want to distinguish between the analog filter and the four following filters, we have rewritten the paragraph following your second recommendation:

L.100-123 : " An initial analog filter is used for filtering erroneous TBs and scanlines. The filter is based on the 16 analog voltage entries in the data. […]

Next, several other filters are employed using the processed TB (in Kelvin) from the previous step. The filters are applied in the following order:

Data that are non-physical and outside the expected range for sea and ice surfaces are removed. Only data points that lie inside the range specified in Eq. 1 are kept: [...]

The next filter removes erroneous scan-lines (across track rows). Consecutive scan-lines should not differ by more than 50K as shown in Eq. 2. The threshold of 50K was estimated experimentally. [...]

n is the maximum across track index of a row, i.e. for a complete row with valid data points for all 78 incidence angles n = 78.

Afterwards, single TB outliers are removed, if they cannot satisfy Eq. 3:[...]

The threshold of 150K was selected manually after identifying erroneous single pixel outliers in the data.

The last filter in Eq. 4 removes neighbouring TBs which are locked on the same value, i.e. TBs for which the following equation equals zero:[...]

where p is again a single pixel TB with an along track index i. The choice of comparing 7 consecutive TBs is based on qualitative experiments. Since the filter is used universally for all incidence angles, the search window varies but covers a minimum distance of 175 km."

103-113: Another issue is that the equations for some of the filters are used to describe included data and some are to described excluded. For example, Eq. 1 shows the range of valid values, while Eq. 2 and Eq. 3 describe thresholds to remove invalid values. It would be better to be consistent and have each equation describe valid values or each describes invalid values. Eq. 4 I'm not quite sure about- does "not equal to zero" mean remove or keep?

Thanks for pointing this out, we have changed the equations and descriptions to be consistent, so that only valid values are described in the equations. Regarding the last filter, if Eq.4 is satisfied, i.e. not equal to zero, the data is kept.

L.112-119: "

$$\frac{\sum_j^n |T_{B_{j,i+1}} - T_{B_{j,i}}|}{n} \leq 50K \tag{2}$$

$$|p_i - p_{i-1}| + |p_{i+1} - p_i| \leq 150K \tag{3}$$

[...]

The last filter in Eq.4 removes neighbouring TBs which are locked on the same value, i.e. TBs for which the following equation equals zero: "

121: Section 3 is labeled as "The radiative transfer model", but the sub-sections seem to go into other areas. I guess the RTM affects the subsections, but it seems like only the first part of Section 3 is specifically RTM and then the subsequent subsections are actually about the

process of deriving SIC. I would suggest putting 3.1 to 3.3 into a Section 4 "Derivation of sea ice concentration", or something along those lines.

The RTM is an essential part of the SIC derivation (the geophysical noise reduction, specifically), as described in 3.1, but we agree that the first part of section 3 is a more general introduction to the RTM, so we have now organized the sections as you propose.

L.131 "3 The radiative transfer model"

L.182-186 "4 Derivation of sea ice concentration

The RTM is an essential part of the SIC derivation for applying an atmospheric noise reduction. The following section presents the calculations of dynamical tie-points and the SIC algorithm, along with uncertainty estimations. Lastly, the post-processing, including land-spill-over method and data flags assignments, is described. A flow-chart illustrating this processing chain can be found in the ESA CCI ESMR product user guide (PUG) (ESA CCI, 2022)."

L.187 "4.1 Tie-points and geophysical noise reduction"

L.241 "4.2 The sea ice concentration (SIC) and its uncertainty"

182: What is the source of the SIC in ERA5? I think this is worth noting since it is an important element in deriving the tiepoints.

We agree that this is important information, and have included it to the tie-point section of the manuscript:

L.195-199 "The ERA5 SIC prior to 1979 is based on the HadISST2.0.0.0 data set (Bell et al. 2021), which mainly utilizes digitized sea ice charts for this period (Rayner_et_al_2003). The two main data sources are the Walsh data set (Walsh, 1978; Walsh and Johnson, 1979; Walsh and Chapman, 2001) and National Ice Center (NIC) charts (Knight, 1984). The data sets also consist of several data types besides ice charts, e.g. ship observations and satellite data, both infrared and microwave observations, including data from ESMR. "

Sources:

[Bell et al. 2021] https://doi.org/10.1002/qj.4174 Describes the ERA5 preliminary back extension and states that the used ERA5 utilizes SIC from the HadISST2.0, which is similar to HadISST1.1 described in [Rayner et al., 2003].

[Rayner et al., 2003] https://doi.org/10.1029/2002JD002670 describes HadISST1 and its data sources, which mainly consists of the digitalized sea ice charts of the Walsh dataset [Walsh, 1978; Walsh and Johnson, 1979; Walsh and Chapman, 2001] and National Ice Center (NIC) charts [Knight, 1984].

[Knight, 1984] https://doi.org/10.3189/1984AoG5-1-81-84 talks about NIC/JIC data, which also uses satellite data in their analysis, both visible/infrared and microwave data, including the ESMR data.

[Walsh, 1978] Walsh, J. E. 1978. A data set on Northern Hemisphere sea ice extent, 1953-76.World Data Center-A for Glaciology, Boulder, Colorado, Glaciological Data Report GD-2, p. 49-51

[Walsh and Johnson, 1979] https://doi.org/10.1175/1520-0485(1979)009%3C0580:AAOASI%3E2.0.CO;2

[Walsh and Chapman, 2001] https://doi.org/10.3189/172756401781818671

184-185: This isn't clear to me. What do you mean by the 15-day averaging period is maintained even at the beginning and end of the data-set and when there are data gaps. I assume this means that wherever there is a gap, valid data would not start until 7 days after the end of that gap so that there is a full 15 days for the average period. Is this correct? If so, maybe slightly rephrasing this to be clear.

We agree that this should be elaborated and have added the clarification to the text. It's good that you raise this point, as it is actually the opposite of how you understood the text, i.e. the 15-day averaging period is maintained, even if there are gaps, the averaging is then done over the days available in the 15-day period. So even in the beginning of the ESMR data-set, where the first 7 days are missing, a valid tie-point is computed. An averaging with a full 15 days only is not possible to maintain, especially for the second half of the dataset, where there is only data available for every other day.

L.204-210 "The 15-day averaging period has been maintained even at the beginning and end of the data-set and when there are data gaps, i.e. if there are gaps, the averaging is done over the days available in the 15-day period, also in the beginning, where the first 7 days are missing.

Figure 2 depicts the 15-day averaged tie-points through time. It shows that the ice tie-points follow a seasonal pattern, while the water tie-points are relatively constant. The tie-point criteria from table 2 ensure that each daily tie-point is based on many observations, which result in stable tie-points. The 15-day interval has been chosen experimentally, so the TB variations seem reasonable and one is still able to identify calibration issues as jumps as e.g. in 1976. "

348-360: I'm not a fan of using bullet points for conclusions in a journal article. Maybe just a personal preference by me, but I think it looks and reads better as paragraphs, especially when the bullet points are complete sentences and not a list.

We value the feedback and have rewritten the conclusions as paragraphs.

L.387-396 "The most important findings can be summarized as:

A comparison to NSIDC's ESMR SIC product and the OSI-SAF CDR was presented. While the seasonal pattern is very similar to NSIDC's ESMR SIC product, our product shows systematic larger SIE values, which can not be explained by differences between land masks alone. For the Northern Hemisphere our SIE values are matching the levels of the 1980s of the OSI-SAF CDR with the same land mask, while values of the Southern Hemisphere have been larger in the 1970s than in the 1980s.

Uncertainty estimates have been included in our ESMR data set for better data assessment and easier comparison to other data sets.

Atmospheric noise has been reduced with the use of an RTM and the ERA5 atmospheric data.

Dynamical tie-points were used to avoid biases from the RTM and NWP data as well as to adjust to seasonal variability and instrument biases."

---

## Author Comment (AC2)

**Reply to Referee Comment RC2**

**(review in black, author response in blue)**

**Review on "Mapping of sea ice concentration using the NASA NIMBUS 5 ESMR microwave radiometer data 1972-1977" by Kolbe et al. (2023)**

*Submitted to Earth System Science Data (ESSD)*
*13 November 2023*
* * *
The submitted paper presents a newly developed sea ice concentration algorithm, that makes use of historic Nimbus-5 ESMR satellite data – a single (low) frequency passive microwave sensor that operated between 1972 and 1977. The new re-processed data set is intended to extend previous CDR of passive microwave sea ice observations back in time. Not only due to an often quite low data quality, with prolonged data gaps especially towards the end of the covered time period, a range of modern processing techniques, various filters steps and additional post-processing are applied to ensure a sufficient output data quality with added uncertainty quantifications. While differences to sea ice extent (SIE) estimations from other comparable OSISAF CDR seem to be rather low, i.e., showing similar magnitudes in the 70ies and 80ies, the authors mnote apparent differences (positive offset) when comparing this new data set to the older NASA ESMR SIC processing by Parkinson et al. (2004). While land mask differences seem to play a role in that context, the overall SIE differences cannot be explained to full extent at this stage, but seem to be more related to algorithm and processing issues.

The paper is nicely written and, in most parts, well-structured and easy to follow. Below, I list several parts that could benefit from some rephrasing/clarification under general and specific comments. Figures and Tables are generally good and informative, but could here and there be improved by some small tweaks, additions and potentially merging (see technical comments below).

Overall, I consider the study well worked out, so that some rather minor changes could well be sufficient to grant publication.
* * *
**General comments**

- I noted several suggestions to make the figures a bit more concise below under "technical comments". While often a stylistic choice, I would consider this to be an area where the paper could easily be improved a bit – also with regard to some descriptions/discussions in sections 4 & 5.
- Please pay attention with your equations - sometimes they miss proper explanations (see specific comments below) and are quite briefly "rushed" over.
- In your "Sea Ice Concentration Product User Guide for ESMR", I noticed a processing flow-chart that didn't make the jump to this paper here. I would consider this to be a nice addition here as well.
- It would have been nice to read a bit more about the core differences between your new approach and the study by Parkinson et al. (2004). For instance, right in the introduction where it is currently just one short sentence. It doesn't need to be overly long, but it might help to get a sense on where algorithm-related differences might originate from, without having to read the full Parkinson et al. study first.

- As already mentioned in RC1, I also got the impression that sub-sections 3.1 to 3.3 do not necessarily relate to a section entitled "The radiative transfer model". Hence, I agree to think about an alternative way of naming / subdividing section 3.

Thank you very much for your useful comments and input. All comments have been accepted (see detailed answers below and the new manuscript). Only the processing flow-chart from the Product User Guide (PUG) has not been added to the paper, instead a reference to the PUG has been added to the manuscript:

L.185/186 " A flow-chart illustrating this processing chain can be found in the ESA CCI ESMR product user guide (PUG) (ESA CCI, 2022)."

Regarding the differences between our approach and (Parkinson et al.,2004): We have clarified in the manuscript, that the following paragraph in the introduction is describing the difference between their study and our methods:

L.46-49 "**Compared to (Parkinson et al., 2004),** this method reduces atmospheric noise regionally over both ice and water surfaces and uses the pre-processed data to develop a SIC algorithm calibration that is effective in removing both instrument drift and offsets. Seasonal sea ice signature variations are removed by using dynamical tie-points. Lastly, the algorithm calculates time and spatially varying uncertainty estimates."

**Specific comments/questions**

**Introduction (Sect.1)**
L.26: I'm sure the references are picked on purpose here, but it seems like a rather long list for this single statement? Is it worth to point out the key differences among those studies here?

The references have been chosen to show that this trend of SIE decline can and has been observed from different observations and data products through time. The chosen references describe data sets of various data sources with different algorithms and trend analyses. We have now separated the dataset and trend analysis references, to better highlight differences between them. The beginning of the introduction has been rewritten to:

L.25-29 "Several sea ice concentration (SIC) algorithms have been developed for passive microwave data (PMW), differing in the usage of e.g. frequencies and polarizations of the PMW data (Comiso et al., 1997), or the usage of static or dynamic tie-points (Parkinson et al., 2004; Tonboe et al., 2016; Lavergne et al., 2019). From the resulting data sets, it becomes apparent that the arctic sea ice extent (SIE) in September has been decreasing at a rate of about 12 percent per decade since the launch of modern satellite multi-frequency microwave radiometers in 1978 (Onarheim et al.,2018; Stroeve and Notz, 2018). "

**Instrument & Data (Sect.2)**

L.56-68: Reference(s) missing for all these platform / sensor specific information.

Thanks for bringing this to our attention, these references have been added.

The technical details have been taken from the Nimbus-5 User Guide (NASA GSFC, 2016) and the ESMR NSIDC Polar Gridded Brightness Temperatures v2 user guide (Parkinson et al., 1999). (https://nsidc.org/sites/default/files/nsidc-0077-v002-userguide.pdf) , for which a new reference has been added. The other information has been acquired from the data itself.

L.58-68: "The NIMBUS 5 ESMR instrument was a cross-track scanner measuring at 78 scan positions perpendicular to the flight track with a maximum incidence angle of about 64 degrees to both sides **(NASA GSFC, 2016)**. No direct observations at nadir have been made, the closest positions being at +/- 0.7 degrees. The near circular orbit height was about 1112 km with an inclination of 81 degrees. The phased array antenna dimensions was 85.5 x 83.3 cm and the spatial resolution about 25 km near nadir increasing to about 160 x 45 km at the edges of the swath **(NASA GSFC, 2016)**. The full swath was about 3100 km with varying incidence angle and spatial resolution giving a very good (unprecedented) daily coverage in polar regions with no gaps, i.e. no pole holes. The ESMR onboard the NIMBUS 5 satellite was a one channel 19.35 GHz horizontally polarised microwave radiometer operating from 11. December 1972 until 16. May 1977 (1617 days) with some interruptions (see list of days with missing files in Appendix A2). Due to a hot-load anomaly, there are major data gaps between March to May and again in August 1973. Another major data gap occurred from 3 June 1975 until 14 September 1975 because the ground segment was used for receiving Nimbus 6 data instead **(Parkinson et al., 1999)**."

L.77: "…and appended to a NetCDF file, …" – it is the same file as the satellite parameters, this could be phrased a bit clearer.

Thanks for pointing this out, it has now been clarified in the manuscript. Yes, the co-located ERA-5 values are added to the raw ESMR NetCDF files, used as input for the SIC processing.

L.78-81 "All variables in the TAP files were read using online NASA software and converted to NetCDF format without changing the original data structure, **creating raw ESMR NetCDF files**. Each data point in the TAP file was matched with European Centre for Medium Range Weather Forecast (ECMWF) ERA5 re-analysis data (**Bell et al., 2020;** Hersbach et al., 2020) in time and space (nearest) and appended to **the raw** ESMR NetCDF file, serving as input to the processing chain."

L.107 (Eq.2): "n" not explained in the following

An explanation has been added:

L.113-114 "n is the maximum across track index of a row, i.e. for a complete row with valid data points for all 78 incident angles n = 78."

L.110 (Eq.3): Was the threshold for Tb outliers (150 K) also experimentally estimated or was it chosen arbitrarily?

L.117-118 "The threshold of 150K was selected manually after identifying erroneous single pixel outliers in the data."

L.111 / Eq.4: There are some explanations for this equation missing (size of the search window & effect of varying it; meaning of ≠ 0)

Thanks, this was also noted by reviewer 1. If Eq.4 is satisfied, i.e. not equal to zero, the data is kept. We revised the filter section and added explanations to the manuscript.

Regarding the size of the search window, the choice of comparing 7 consecutive TBs is based on qualitative experiments, after obviously corrupted data was found by plotting the raw swath data. The search window shall ensure that the filter only finds points where the observations are corrupted, so that the values do not change at all for consecutive points in

the along track direction. All decimal values are taken into account, though the ESMR TBs are only provided with one decimal, so several points are needed to cover a long enough distance to make sure that the constant TB value is not caused by the observed surface. We would expect the TBs to at least vary slightly for points that are a few hundred kilometres apart. Since this filter is used universally for all incidence angles the smallest distance occurs at the incidence angles closest to nadir, where a sufficient number of neighbouring points are needed to make sure that the non-variant TBs are not natural.

L.121-123 "The choice of comparing 7 consecutive TBs is based on qualitative experiments. Since the filter is used universally for all incidence angles, the search window varies but covers a minimum distance of 175 km."

**Radiative transfer model (Sect.3)**

L.145 (Eq.7): Careful with the second use of "p". Although different from earlier uses as "p_i" as single pixel indicators, it might be confusing for the reader and should, for instance, be augmented by an additional explanation of "p" and "q" in Eq.7.

Thanks for pointing this out, we added a clarification that p & q are abbreviations in equations 6-8:

L.153 "where p and q are abbreviations for:"

L.182: Please provide more details on the ERA5 OSTIA SIC, as they are quite crucial for the selection of tie points (affecting SIC thresholds and presumably also distance to the ice edge). Further, how is the distance to the ice edge defined, and where/how is this criterion depicted in Table 2?

Thank you for bringing this to our attention, this was also noticed by reviewer 1. ERA5s SIC prior to 1979 comes from the Met Office's HadISST2.0.0.0 product. We agree that this is important information, and have included it to the tie-point section of the manuscript:

L.195-199 " The ERA5 SIC prior to 1979 is based on the HadISST2.0.0.0 data set (Bell et al. 2021), which mainly utilizes digitized sea ice charts for this period (Rayner_et_al_2003). The two main data sources are the Walsh data set (Walsh, 1978; Walsh and Johnson, 1979; Walsh and Chapman, 2001) and National Ice Center (NIC) charts (Knight, 1984). The data sets also consist of several data types besides ice charts, e.g. ship observations and satellite data, both infrared and microwave observations, including data from ESMR. "

Sources:

[Bell et al. 2021] https://doi.org/10.1002/qj.4174 Describes the ERA5 preliminary back extension and states that the used ERA5 utilizes SIC from the HadISST2.0, which is similar to HadISST1.1 described in [Rayner et al., 2003].

[Rayner et al., 2003] https://doi.org/10.1029/2002JD002670 describes HadISST1 and its data sources, which mainly consists of the digitalized sea ice charts of the Walsh dataset [Walsh, 1978; Walsh and Johnson, 1979; Walsh and Chapman, 2001] and National Ice Center (NIC) charts [Knight, 1984].

[Knight, 1984] https://doi.org/10.3189/1984AoG5-1-81-84 talks about NIC/JIC data, which also uses satellite data in their analysis, both visible/infrared and microwave data, including the ESMR data.

[Walsh, 1978] Walsh, J. E. 1978. A data set on Northern Hemisphere sea ice extent, 1953-76.World Data Center-A for Glaciology, Boulder, Colorado, Glaciological Data Report GD-2, p. 49-51

[Walsh and Johnson, 1979] https://doi.org/10.1175/1520-0485(1979)009%3C0580:AAOASI%3E2.0.CO;2

[Walsh and Chapman, 2001] https://doi.org/10.3189/172756401781818671

Regarding the distance to the ice edge criteria: This is not imposed as a distance threshold, but as the "mean sea ice concentration (ERA5) of a 5 x 5 grid point box" shown in table 2, which for ice should be >0.8 and for water tie points <0.01 .

We have added a clarification to the manuscript:

L.201-202 "The distance from ice edge criterium is imposed by putting a threshold on the mean SIC of a 5 by 5 grid point box, which for ice tie points should be larger than 80% or less than 1% for open water points. "

Fig.3: Add data source to caption.

We added ERA5 as the data source for the water vapor to the caption:

Fig.3: "Water vapor data from ERA5 (Hersbach et al.,2020)."

L.200: Add data source for T2m data.

ERA5 was added as the data source in the text:

L.225 "where T2m is the 2 m air temperature, which is taken from the ERA5 data."

**Results (Sect.4)**

L.267: "This is a consequence of the one channel SIC algorithm." – Can you comment more on this effect? Is there any way / idea to correct for this?

Single channel SIC algorithms have an inherent ambiguity between SIC, ice type and temperature variations. While a single frequency can be enough to detect the difference in surface emissions caused by different physical properties of sea ice, it is challenging to accurately distinguish whether these are caused by differences in ice type, temperature or SIC. This results in e.g. the underestimation of the multiyear ice (MYI) SIC.The focus in this first version of the dataset was on the first year ice (FYI) and marginal ice zone, to be able to estimate a sea ice extent. While the SIC of the MYI is underestimated, it is still above the threshold for the sea ice extent. To improve the SIC of the multiyear ice, additional information like the sea ice type are necessary to improve the algorithm, so that e.g. tie-points can be calculated for FYI and MYI individually, to improve the calculated SIC.

L.292-294 " This is a consequence of the one channel SIC algorithm, which has an inherent ambiguity between SIC, ice type and emitting layer temperature variations. "

L.267/268: Uncertainties are also high along the coastal margins, especially at lower latitudes. Might be worth to mention and explain.

This is a good point and has been added to the manuscript. Coastal regions show also higher uncertainties for the same reason as around ice edges, since the land-spill-over correction is first applied to the SIC after the uncertainty estimations.

L.295-297: "Coastal regions also show higher uncertainties for this reason, since the land-spill-over correction is first applied to the SIC after the uncertainty estimations."

L.270-272: As noted under technical comments below - hard to locate for readers that are unfamiliar with the geographical setting of one or the other hemisphere. Simple lat/lon indications could help (when present on the maps), or alternatively, a regional close-up figure (daily level? Could then even feature other parameters from the data set, such as uncorrected/corrected brightness temperatures).

Thanks for pointing this out, we have added lat/lon indications to all map plots and added coordinates to the text, as well as some references to papers describing these areas.

L.298-301 "One such feature is the Odden ice tongue **(Comiso et al., 2001)** extending eastward from the East Greenland Current, visible in figure 6 **(around 73°N, 0°E)**, while another feature is the Maud Rise Polynya **(Jena et al., 2019)**, an open water area encircled by sea ice, in the Southern Hemisphere, which can be seen in figure 7 **(around 65°S, 0°E**)."

L288: "For the Northern Hemisphere the SIE seems to have been slightly lower during the operational period of NIMBUS 5 ESMR 1972 to 1977 than during the operational period of NIMBUS 7 SMMR from 1978 to 1987."
– To me & purely based on Fig.14, max. values in winter seem to be more or less similar, while the two available min. values in summer are seemingly even slightly higher than in the 80ies. Am I wrong?

Thank you very much for bringing this to our attention, the description has been corrected:

L.316-318 "For the Northern Hemisphere the SIE seems to have been similar in magnitude during the operational period of NIMBUS 5 ESMR 1972 to 1977 as during the operational period of NIMBUS 7 SMMR from 1978 to 1987, with the ESMR minimum extents being slightly higher than the SMMR ones."

**Discussions (Sect.5)**

L.300: Is it possible to illustrate this with an example? I.e., where do the land masks differ the most?

Thanks for bringing this up, as we previously hadn't investigated the different NSIDC land masks further, since no ESMR land mask file is available. We contacted NSIDC, who haven't made a specific comparison between the ESMR and CDR land masks themselves, but they can report that the source data is the same for both of them and there might be some small differences due to the re-gridding process.

A special report has been published last year, focusing on the evolution of NSIDC's land/surface masks: https://nsidc.org/sites/default/files/documents/technical-reference/nsidc-special-report-25.pdf

Consequently, we have taken out the statement about the NSIDC land masks and rewritten the paragraph:

L.321-327 "Comparisons between different sea ice products and the new ESMR data set proved to be more difficult than initially expected, since not only the processing algorithms

differ, but also the land masks**, map projections and data set grids**. We were not able to find two independent SIC data sets for 1978 onwards and 1972-77, which share **exactly** the same land mask.

Thus, it was decided at the beginning of the processing to use the same land mask as the OSI-420 product (1978 onwards) (EUMETSAT, 2020) for our ESMR data set, i.e. a 25 km equal area grid (EASE-2 version 2) land mask, to at least ensure a fair comparison between these two data sets. The NSIDC ESMR data set (green line in figure **9**) used a different land mask with a polar stereographic projection (Parkinson et al., 2004)**, but was still compared to our ESMR data set.** "

L.325: You mention a planned reprocessing of some sort to increase the number of rescued data points. Do you already have concrete ideas on how you intend to do that?

There are two approaches to improve the data selection during the next processing.

One approach is to change the set of filters, which have been used to remove erroneous data. The filters used in this first time processing have been very strict and used universally (e.g. for all incidence angles in the same way). Therefore, an adjustment of the filter thresholds, usage of different filters, or incorporation of incidence angle dependency could improve the data selection.

Another approach is related to the calibration of the data. Systematic erroneous data has been identified, where whole swaths of data could potentially be rescued by a re-calibration with a new look-up table across the swath.

We added the following to the manuscript:

L.352-356 "There are two approaches to improve the data selection during the next processing. One approach consists of changes to the set of filters, i.e. an adjustment of the filter thresholds, testing of different filters and a possible incorporation of an incidence angle dependency to the data selection.

The other approach is to recalibrate some of the erroneous data files, which have shown some systematic offsets, to rescue whole swaths. "

L.330: Can you name some examples for "related snow and ice processes" that are causing the noise over sea ice? How about atmospheric effects over sea ice that also trouble other widely used SIC algorithms?

The measured brightness temperature  can be influenced by sea ice deformation, leads or ridges of the ice. Additionally, snow cover can have an impact as well (which varies depending on the snow depth, snow density and grain size) There can also be meltwater within the snow layer, which influences scattering and emission processes inside the layer. Especially during summer melt and refreezing processes can therefore result in noise of TB measurements.

Atmospheric noise caused by e.g. water vapor and cloud liquid water exists also influences the TB over sea ice, but its influence is smaller (around ⅓ of the total noise [Tonboe et al.,2021]) compared to the geophysical noise from the snow and ice properties.

[Tonboe et al., 2021] Tonboe, R., Nandan, V., Mäkynen, M., Pedersen, L., Kern, S., Lavergne, T., Øelund, J., Dybkjær, G., Saldo, R., and Huntemann, M.: Simulated Geophysical Noise in Sea Ice Concentration Estimates of Open Water and Snow-Covered

Sea Ice. IEEE Journal of Selected Topics in Applied Earth Observations and Remote Sensing. PP. 1-1. 10.1109/JSTARS.2021.3134021, 2021.

We added the following to the manuscript:

L.361-365 "Such processes include sea ice deformation, creation of leads or ridges, as well as changes in the snow layer of e.g. snow depth, snow density and, grain size, but also melting and refreezing, which influences scattering and emission processes inside the layer. Atmospheric noise caused by e.g. water vapor and cloud liquid water influences the TB over sea ice, but its influence is smaller, only around one third of the total noise (Tonboe et al., 2021)."

**Technical comments**

L.82: Brackets around reference missing (NASA CR, 1974)

The brackets around the reference have been added. (now L.85)

L.100: "a value of 10" – unit missing

We would like to add units, but this is referring to the analog signals of the raw ESMR data, for which the ESMR Nimbus-5 User guide does not provide units and only calls them "voltages of engineering interest" (page 83, https://ntrs.nasa.gov/api/citations/19740020209/downloads/19740020209.pdf ).

We therefore added the following to the manuscript:

L.101/102 "The **NIMBUS 5 ESMR** user's guide **(Sabatini, 1972)** does not explain very well what the 16 entries really are**, or what unit the voltages are stored in,** but jumps in these analog signals correspond to anomalous TBs. "

Figures 2, 3, 4 & 5: Similar to other line plots, the addition of grid lines could help to make out differences more easily.

Grid lines have been added to these figures.

Figure 4 & 5: Smaller points could help to reduce the large red "cluster-patches". Further, the captions read more like notes, this could be improved by using proper phrases. Lastly, Fig.4 & 5 could be combined into one Figure 4 (a & b), just to spare two almost identical captions next to each other.

Thanks for the suggestions, the figures have been updated accordingly.  Due to the smaller points the individual lines corresponding to the different incidence angles (scan positions) have also become visible. Point size and opacity have been tuned, there is still some cluttering, but even smaller sizes would make the less populated areas more difficult to read (this is of course up to personal preference). The cluttering around the zero line remains and is something the plot is intended to show.

Figure 6 & 7: As before - could be combined into one Figure (a & b) to spare two almost identical captions next to each other.

Figure 6 & 7 have been combined into one Figure 5 (a & b).

L.219-225: Steps 1) and 2) could be formatted as bullet points, thereby reducing potential confusions with the numbers just before point (2).

The text has been re-formatted.

L.246-252 "

1) The c_ice is first estimated using uncorrected TB s and tie-points derived from uncorrected data. The c_ice estimate is truncated to the interval between 0 and 1 and an open water filter is applied, forcing all c_ice values less than 0.15 to 0.

2) The c_ice estimate from step (1) is used in the RTM calculation (Eq. 5) together with ERA5 data for the geophysical noise reduction of the TBs and c_ice is then estimated again in a second iteration, this time using corrected TBs and corrected tie-points. The mean values of V, W, L... used in the reference simulation is a weighted average with c_ice of the mean water and ice tie-point values respectively, i.e. c_ice is used as a ratio to mix the two tie-point values to create mean values of the NWP data for any sea ice concentration."

L.254-255: Double use of "also".

The first "also" was deleted. (now L.281/282)

Figures 8-13: Multiple comments/suggestions

- Size of individual panels could be increased
- Geographic references are missing in all panels – e.g., thin lat/lon lines could be added
- Is the wide spatial extent of the maps chosen on purpose or would it make sense to zoom in a bit, sparing some lower latitude regions?
- Monthly mean SIC & uncertainties could be combined in merged Figures (a & b)

Thank you for your suggestions, we adjusted all figures following your advice. Regarding the wide spatial extent, this was chosen on purpose to show the large extent of the dataset and to act as geographic reference for experienced readers. Since we added lat/lon lines now, we have zoomed in slightly, but are still showing most of the dataset.

L.280 & 283: "threshold of 30%" – better write s.th. like "c_ice ≥ 30%"

The threshold is now being referred to in the text as: "a threshold of c_ice > 30%" in L.299 & L.301/302.

Figure 14 & 15: As before - could be combined into one Figure (a & b) to spare two almost identical captions next to each other. Plus: be more specific with the thresholds: "… a 30% sea ice concentration (SIC) threshold…" and further "…, where the 30% SIC-threshold has been applied".

The figures have been combined into one figure, with changes as in L. 308 & L. 301/302 in the caption.

L.308 "...mean SIE are calculated using a threshold of c_ice > 30%"

L. 311-312 "...using the same threshold for all products (c_ice > 30%)."

L.348 / Section 6: reads more like a Summary and is quite short. One could think about merging this part with section 5 as a "Discussions & summary" chapter, but this is certainly personal taste.

Thanks for the suggestion. We changed section 6 in response to referee response #1, and therefore we chose not to merge it with the discussion section in this revision.

Table A1 (Appendix A): Under "ERA5 variables", I think there is an error with u10m / v10m and their latitude / longitude reference.

Thanks for bringing this to our attention, this has now been corrected.

Table A1:

u10 - u component of the wind speed at 10 m (parallel to **latitude**) [ms−1]

v10 - v component of the wind speed at 10 m (parallel to longitude) [ms−1]

Dataset entry on CEDA archive: The document "Algorithm Theoretical Basis Document (ATBD) - ESMR Sea Ice Concentration" is currently not accessible from the ESA website. Has it been moved?

Thank you for pointing this out, this was an error on the homepage and it has been fixed.

The ATBD (and other ESMR documents) are also available through the ESA Sea Ice webpage: https://climate.esa.int/en/projects/sea-ice/Sea-Ice-Key-Documents/

---

## Author Response (AR2)

**Author response to editor comments**

**(editor comments in black, author response in blue, all line numbers refer to the ones in the updated manuscript)**

**Editor comments:**

Thank you for your comprehensive response to almost all reviewer comments. I did an editorial pass for typos/other grammatical things: All line numbers refer to the new document. Following these changes, I will be happy to accept the article for publication.

3: Change period after 16 to a comma

L.3 "[...] May. 16**,** 1977."

18: replace periods with commas in the 6-digit numbers

L.18 "[...] differences of 240**,**000 and 590**,**000 km², respectively."

A typo on line 28 (arctic without capital A)

L.28 "[...] the **A**rctic sea ice extent [...]"

An error has been introduced on new Line 149 in the Wentz reference format.

Thanks for pointing this out, we assume this is a typo and refers to new line 139, where we now have removed the brackets.

L.139 "The RTM uses the atmospheric part of the model described in **Wentz, 1997** to compute [...]"

Same on new line 167.

L.167 "[...] described in **Wentz, 1997**."

201: Need a space before the start of the new sentence

L.201 "[...] in table 2. The distance [...]"

387 - 397: The suggestion from Reviewer 1 to remove the bullet points was a good one -- but I think your response isn't quite sufficient here. This list is still basically in bullet point format, but with the bullet points removed. I suggest that you rewrite this into one or two paragraphs -- not a series on single sentence, disjointed paragraphs.

Thank you for your time and effort put into reviewing this paper. We made the suggested changes to the final version of our manuscript as stated above and have rewritten the conclusion:

L.383-398 "

In this paper we presented a new SIC data set covering 1972-1977, by using the ESMR data from the Nimbus-5 satellite. The data set consists of resampled daily netCDF files for the Northern and Southern Hemispheres, respectively. SIC, associated uncertainties and processing flags are included in the data set. The uncertainties follow the same principles as the ones of the EUMETSAT SIC CDR, including both algorithm and re-sampling uncertainties. The choice of same land mask, spatial grid and projection as for EUMETSATs SIC CDR make comparisons between the time periods easier.

A comparison to NSIDC's ESMR SIC product and the OSI-SAF CDR showed that the seasonal pattern is very similar to NSIDC's ESMR SIC product, but our product shows systematically larger SIE values, which can not be explained by differences between land masks alone. For the Northern Hemisphere our SIE values are matching the levels of the 1980s of the OSI-SAF CDR with the same land mask, while values of the Southern Hemisphere have been larger in the 1970s than in the 1980s.

Compatibility with the EUMETSATs SIC CDR was achieved by using a similar processing chain. The processing included an atmospheric noise reduction with the use of an RTM and the ERA5 atmospheric data, which lowered the standard deviation of the TBs consistently. Additionally, dynamical tie-points were used to avoid biases from the RTM and NWP data as well as to adjust for seasonal variability and instrument biases. To ensure better data assessment in analysis and in models and easier comparison to other data sets, temporally and regionally varying uncertainty estimates have been included in our ESMR data set. "